# Frechet-power function distribution:Theory, properties and applications

**Merga Abdissa Aga** ⓘ *

Department of Statistics, Salale University, Fiche, Oromia Region, Ethiopia

* mergabdisa3@gmail.com

## Abstract

We propose the Fréchet–Power Function (FPF) distribution, a novel two-parameter model that combines the bounded support of the Power Function distribution with the heavy-tailed flexibility of a Fréchet-type generator. This combination enables the FPF to capture complex features such as skewness, heavy tails, and diverse hazard rate shapes—limitations present in existing bounded lifetime models. We provide explicit forms for the probability density, cumulative distribution, and quantile functions, along with detailed statistical properties including moments and hazard rate behavior. Parameters are estimated using maximum likelihood, with bootstrap and simulation techniques employed to assess estimator performance. Empirical applications to survival, reliability, and environmental datasets show that the FPF distribution consistently outperforms traditional models in terms of goodness-of-fit and flexibility. This work introduces a powerful and versatile tool for modeling bounded lifetime data, offering enhanced accuracy and interpretability across disciplines.

## 1. Introduction

Modeling complex data that exhibit bounded support, asymmetry, and heavy tails remains a significant challenge in modern statistical theory and practice. Such data are commonly encountered in reliability engineering, environmental science, economics, and biomedical research [1–4]. While many classical distributions—like the Power Function—are widely used for modeling bounded data due to their simplicity and interpretability [5–9]. However, they often fail to capture critical features such as heavy-tailed behavior and flexible hazard rate shapes, which are particularly important in reliability and survival contexts.

To overcome these shortcomings, several extensions of the Power Function distribution have been proposed, including the Exponentiated Power Function [10], Weibull-G Power Function [11], Kumaraswamy–Power Function [12], Beta Power Function [13], exponential power [14] and Transmuted Power Function [15,16]. These approaches introduce additional parameters or generator mechanisms to improve

**Data availability statement:** All relevant data are within the paper.

**Funding:** The author(s) received no specific funding for this work.

**Competing interests:** The authors have declared that no competing interests exist.

flexibility. Yet, most of them either cannot adequately model heavy tails or are defined on unbounded supports, limiting their use when data are naturally bounded.

On the other hand, heavy-tailed models such as the Fréchet distribution are well-established in extreme value theory and are effective in modeling data with significant tail risk [17–18]. However, the Fréchet and its related families such as Slash Exponential Fréchet [19], Odd Fréchet-G [20–21] and the Incomplete-Gamma Fréchet [22], Topp-Leone Kumaraswamy Fréchet distribution [23], and A hybrid cosine inverse Lomax-G family of distributions [24], as well as bounded variants like the Zero-Truncated Poisson–Power Function distribution [25], further enrich modeling choices. Nevertheless, these distributions are generally defined on unbounded domains, making them unsuitable when observations are inherently restricted to a finite interval.

Generator methods—which compound a baseline distribution with a flexible generator function—have shown promise for building flexible families of distributions such as: Beta-G [26], Kumaraswamy-G [27], Marshall–Olkin-G [28]. However, to date, no generator framework has combined a bounded baseline distribution with a modified Fréchet generator to obtain a bounded distribution that also exhibits heavy tails. The motivation for employing the modified Fréchet generator lies in its ability to capture complex tail behaviors and diverse hazard rate patterns, features often observed in survival and reliability studies where extreme outcomes play a critical role.

This study proposes the Fréchet-Power Function (FPF) distribution, a novel bounded continuous distribution that bridges the gap between existing bounded Power Function models and heavy-tailed Fréchet-type distributions. By integrating a modified Fréchet generator with a Power Function baseline, the FPF uniquely combines bounded support with flexible tail behavior, controlled by an additional shape parameter. This innovation allows more accurate modeling of skewed and heavy-tailed bounded data commonly encountered in reliability, survival, and environmental applications. The objectives are to establish the distribution's mathematical foundations, derive its statistical properties, develop estimation methods, and illustrate its superior performance through real-data applications.

By explicitly bridging the gap between bounded-only models and heavy-tailed unbounded families, the FPF distribution fills an important niche in the statistical literature. To emphasize its novelty, we provide a comparative summary (Table 1) contrasting bounded, heavy-tailed, and hybrid models, which clearly highlights the unique role of the proposed FPF distribution.

Table 1 clearly shows that most existing distributions are either constrained to monotone hazard shapes (e.g., Power Function, Exponential). By contrast, the proposed FPF distribution uniquely integrates bounded support with flexible tail behavior—capable of accommodating light, heavy, or bounded tails—while also allowing full hazard-rate flexibility (increasing, decreasing, bathtub-shaped, and unimodal forms). This novel combination makes the FPF distribution particularly well-suited for modeling complex lifetime, reliability, and survival data where both finite boundaries and extreme outcomes must be properly captured.

**Table 1. Comparative summary of competing distributions (hazard shapes).**

| Distribution | Support | Hazard Shape | Parameters | Notes |
|---|---|---|---|---|
| α-Power Transformed Power Function | $0 < x < 1$ | Monotone increasing or decreasing | $\alpha, \theta$ | Limited flexibility |
| Exponential | $x \geq 0$ | Constant hazard | $\lambda$ | Memoryless property |
| Power Function | $0 < x < 1$ | Monotone increasing or decreasing | $\theta$ | Simple, classical model |
| Weibull Power Function | $0 < x < b$ | Increasing, decreasing, or bathtub-like | $\alpha, \theta, b$ | Flexible hazard forms |
| Kumaraswamy Power Function | $0 < x < b$ | Monotone or unimodal | $\alpha, \beta, \theta, b$ | Limited hazard flexibility |
| Marshall–Olkin Power Function | $0 < x < b$ | Increasing | $\alpha, \beta, b$ | Adds shock parameter |
| Exponentiated Power Function | $0 < x < 1$ | Monotone increasing | $\alpha, \theta$ | Extension of PF, limited tail control |
| Beta–Power Function | $0 < x < 1$ | Monotone or unimodal | $\alpha, \beta, \theta$ | Flexible shape, bounded support |
| Transmuted Power Function | $0 < x < 1$ | Monotone increasing or decreasing | $\alpha, \theta, \lambda$ | Hazard flexibility, bound |
| Proposed FPF (This Study) | $a < x < b$ | Fully flexible: increasing, decreasing, bathtub, unimodal | $\alpha, \theta$ | Only bounded model combining flexible hazard and tail behaviors |

## 2. The proposed Fréchet Power Function (FPF) distribution

Let X be a continuous random variable with support $(a, b)$, where $a < b$ are known constants. he baseline Power Function distribution has cumulative distribution function (CDF) and probability density function (PDF):

$$F_0(x) = \left(\frac{x-a}{b-a}\right)^\theta, \quad a < x < b, \quad \theta > 0 \tag{1}$$

$$f_0(x) = \frac{\theta}{b-a}\left(\frac{x-a}{b-a}\right)^{\theta-1}, \quad a < x < b, \quad \theta > 0 \tag{2}$$

The Power Function is widely used to model bounded data with skewness [6,29]. The Fréchet distribution, a member of the extreme value family, is known for modeling heavy-tailed phenomena over unbounded support $(0, \infty)$. To construct a flexible family combining bounded support and heavy-tail behavior, we apply a modified Fréchet generator to the Power Function baseline distribution.

For $\alpha > 0$, define the Fréchet Power Function CDF as [30]:

$$F(x) = \frac{e^{-(F_0(x))^\alpha} - 1}{e^{-1} - 1} \tag{3}$$

Differentiating with respect to x gives the PDF:

$$f(x) = \frac{\alpha f_0(x)[F_0(x)]^{\alpha-1} e^{-(F_0(x))^\alpha}}{1 - e^{-1}}, \quad a < x < b \tag{4}$$

Substituting $F_0(x)$ from equation (1), the FPF PDF can be written as:

$$f(x; \theta, \alpha) = \frac{\alpha\theta}{(b-a)(1-e^{-1})}\left(\frac{x-a}{b-a}\right)^{\theta\alpha-1}\exp\left\{-\left(\frac{x-a}{b-a}\right)^{\theta\alpha}\right\}, \quad a < x < b \tag{5}$$

Where $a < b$: Support boundaries

$\quad \theta > 0 \quad$: Baseline Power Function shape parameter

$\quad \alpha > 0$: Fréchet generator additional shape parameter

**Theorem1:**.The FPF distribution $f(x; \theta, \alpha)$ is a legitimate probability density function.

**Proof:** To verify that $f(x; \theta, \alpha)$ is a valid PDF, we must show two things:

a. Non-negativity: For all $x \in (a, b)$, $f(x; \theta, \alpha) \geq 0$.

b. The integral of the PDF over the support $(a, b)$ must be equal to 1:

$$\int_a^b f(x; \theta, \alpha) dx = 1$$

Substitution: Let $z = \left(\frac{x-a}{b-a}\right)^{\theta\alpha} \rightarrow x = a + (b-a)z^{\frac{1}{\theta\alpha}}, \qquad dx = \frac{(b-a)}{\theta\alpha} z^{(\frac{1}{\theta\alpha})-1} dz$

Changing variables in the integral:

$$\int_a^b f(x; \theta, \alpha) dx = \int_0^1 \frac{\theta\alpha}{1-e^{-1}} z^{\theta\alpha-1} \exp(z) dz = \left(\frac{\theta\alpha}{1-e^{-1}}\right)\left(\frac{1-e^{-1}}{\theta\alpha}\right) = 1$$

Thus, $f(x; \theta, \alpha)$ is a legitimate probability density function.

## 3. Statistical properties of FPF distribution

### 3.1. Survival function and hazard function

In survival and reliability analysis, the survival function and hazard rate function play a key role in describing the lifetime behavior of a system or component. The survival function, S(x), represents the probability that a unit will survive beyond time xxx, while the hazard function, h(x), describes the instantaneous rate of failure at time x given survival up to that time. These functions are essential for understanding the risk dynamics of lifetime data and for comparing competing reliability models.

The survival function is: $S(x) = 1 - F(x)$

$$S(x) = \frac{1 - \exp\left\{-\left(\frac{x-a}{b-a}\right)^{\theta\alpha}\right\}}{1 - e^{-1}}, \quad a < x < b \tag{6}$$

**3.1.1. Hazard rate function.** The hazard function is defined as: $h(x) = \frac{f(x)}{S(x)} = \frac{\frac{\theta\alpha}{(b-a)}\left(\frac{x-a}{b-a}\right)^{\theta\alpha-1} exp\left\{-\left(\frac{x-a}{b-a}\right)^{\theta\alpha}\right\}}{1 - exp\left\{-\left(\frac{x-a}{b-a}\right)^{\theta\alpha}\right\}}$

So the final expression is:

$$h(x) = \frac{\theta\alpha}{b-a}\left(\frac{x-a}{b-a}\right)^{\theta\alpha-1} \cdot \left(\frac{exp\left\{-\left(\frac{x-a}{b-a}\right)^{\theta\alpha}\right\}}{1 - exp\left\{-\left(\frac{x-a}{b-a}\right)^{\theta\alpha}\right\}}\right), \quad a < x < b \tag{7}$$

This hazard function is capable of modeling increasing, decreasing, or bathtub-shaped hazard rates depending on the values of $\theta$ and $\alpha$, making the FPF distribution particularly suitable for reliability and survival applications.

Fig 1 shows the hazard-shape parameter map of the FPF distribution over a grid of α and θ values. Blue regions correspond to increasing hazards, red regions to decreasing hazards, and green regions to bathtub-shaped hazards. The map illustrates the distribution's flexibility in modeling diverse hazard patterns and highlights parameter regions where hazards

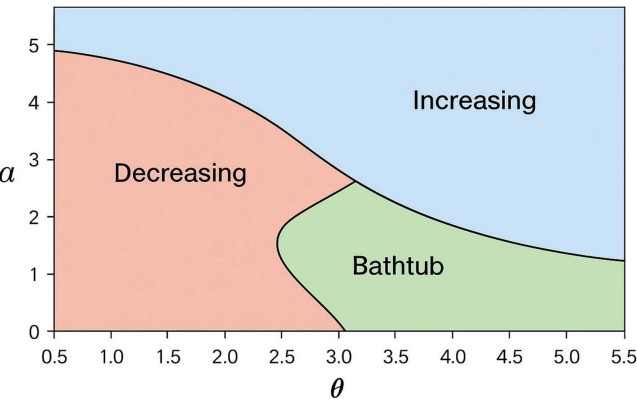

**Fig 1. Hazard-shape map of the FPF distribution over α and θ values.**

may be similar, indicating potential non-identifiability. Notably, strictly increasing hazards occur only at high α and θ values, while decreasing and bathtub-shaped hazards dominate lower parameter ranges.

### 3.2. Plots of the FPF distribution

Figs 2–4 illustrate the shape flexibility of the Fréchet–Power Function (FPF) distribution, showcasing its probability density function (PDF), cumulative distribution function (CDF), survivl function, and hazard rate behavior under various parameter settings.

The product $\theta\alpha$ plays a key role in determining the distribution's characteristics. When $\theta\alpha < 1$, the distribution has a monotonically increasing density and a J-shaped cumulative distribution function (CDF), representing data where observations concentrate near the upper bound. This behavior is appropriate for modeling late-occurring events, such as component lifetimes with early reliability and sudden failure near the end of a cycle (Fig 2). When $\theta\alpha > 1$, the distribution becomes unimodal, producing an increasing-then-decreasing hazard function. The survival function, which represents the probability of surviving beyond a specific point x, starts near 1 when x is close to the lower bound *a* and monotonically decreases to 0 as x approaches the upper bound *b*. The steepness of this decline is governed by the product $\theta\alpha$; larger values lead to a rapid drop in survival probability, indicating a system that fails quickly, whereas smaller values result in a slower decline, suggesting a more resilient or long-lasting system (Fig 3). The hazard function, which indicates the instantaneous failure rate given survival up to time x, also depends critically on $\theta\alpha$. When $\theta\alpha > 1$, the hazard function increases with time, reflecting aging systems with escalating risk. In contrast, when $\theta\alpha < 1$, the hazard function decreases, indicating a high initial risk that lessens over time, characteristic of early-life failures or infant mortality behavior. When $\theta\alpha$ is approximately 1, the hazard rate tends to remain nearly constant, suggesting a random failure process without a time-dependent trend (Fig 4). In some parameter combinations, the hazard function can also display non-monotonic behavior, such as a hump-shaped curve, representing systems that initially experience increasing risk followed by a decline. These features make the FPF distribution suitable for a wide range of practical applications, including survival analysis, bounded time-to-event modeling, reliability and risk studies, environmental modeling (e.g., rainfall within thresholds), and economics, where variables like proportions, bounded growth, or index values must remain within fixed limits. The FPF model offers a valuable extension to the family of bounded distributions by allowing both skewness and flexibility in hazard structure, thereby providing a realistic and adaptable framework for modeling data with constrained support and non-uniform risk patterns.

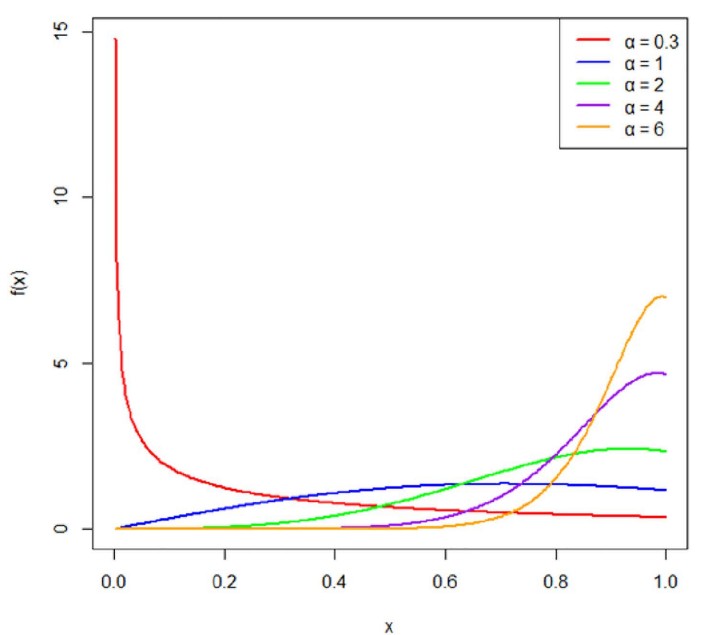

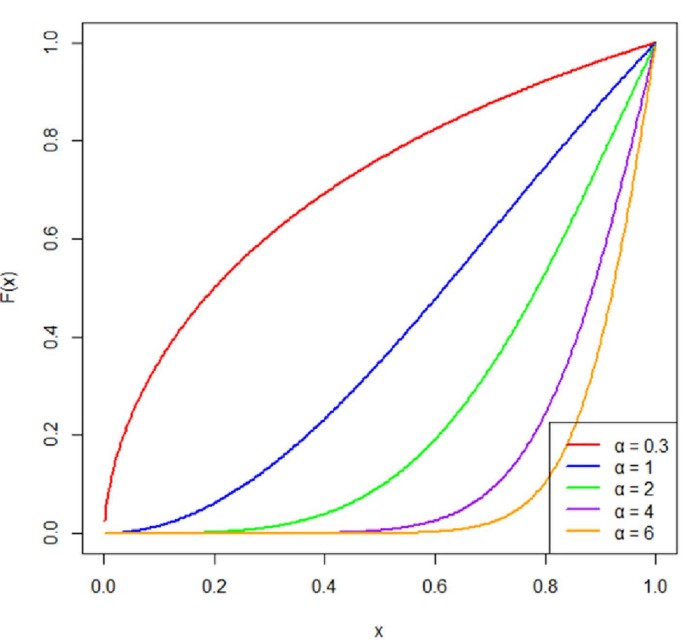

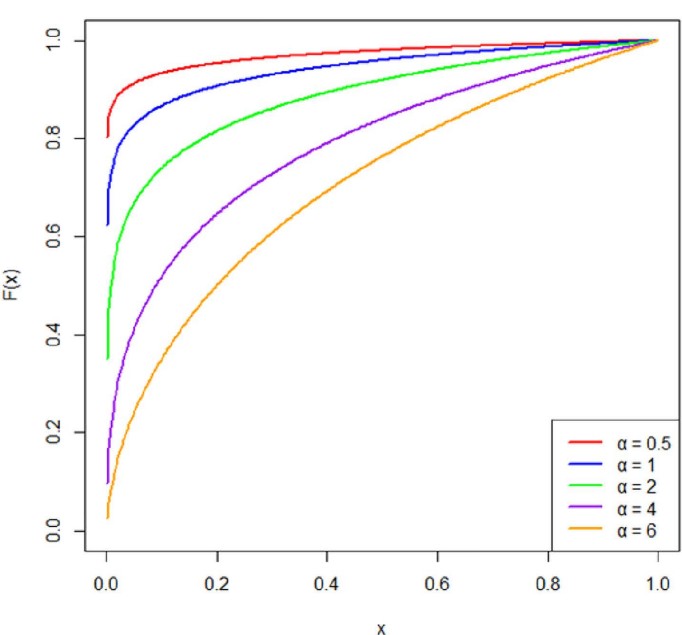

**Fig 2. Probability density function (PDF) of the Fréchet Power Function (FPF) distribution for** $\theta = 2$ **with selected values of** $\alpha$ (**0.5**, **1**, **2**, **4**, **6**), **and cumulative distribution function (CDF) plots for** $\theta = 0.1$ **and** $\theta = 2$ **with the same α values.**

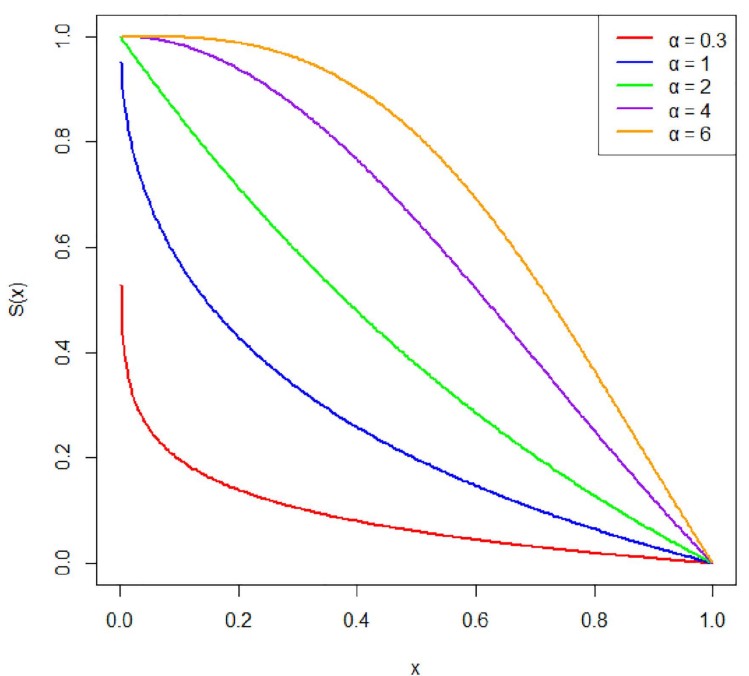

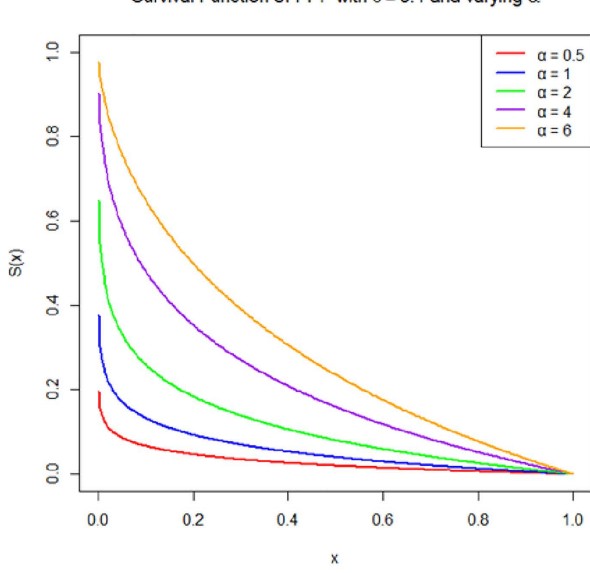

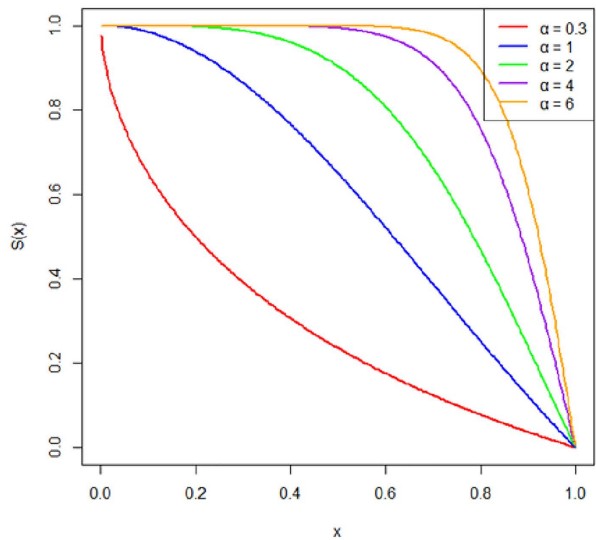

**Fig 3. Estimated survival functions of the Fréchet Power Function (FPF) distribution.** Each panel corresponds to a fixed baseline shape parameter θ (0.1, 0.5, 2). Within each panel, survival curves are shown for generating shape parameters $\alpha = 0.3, 1, 2, 4,$ and 6.

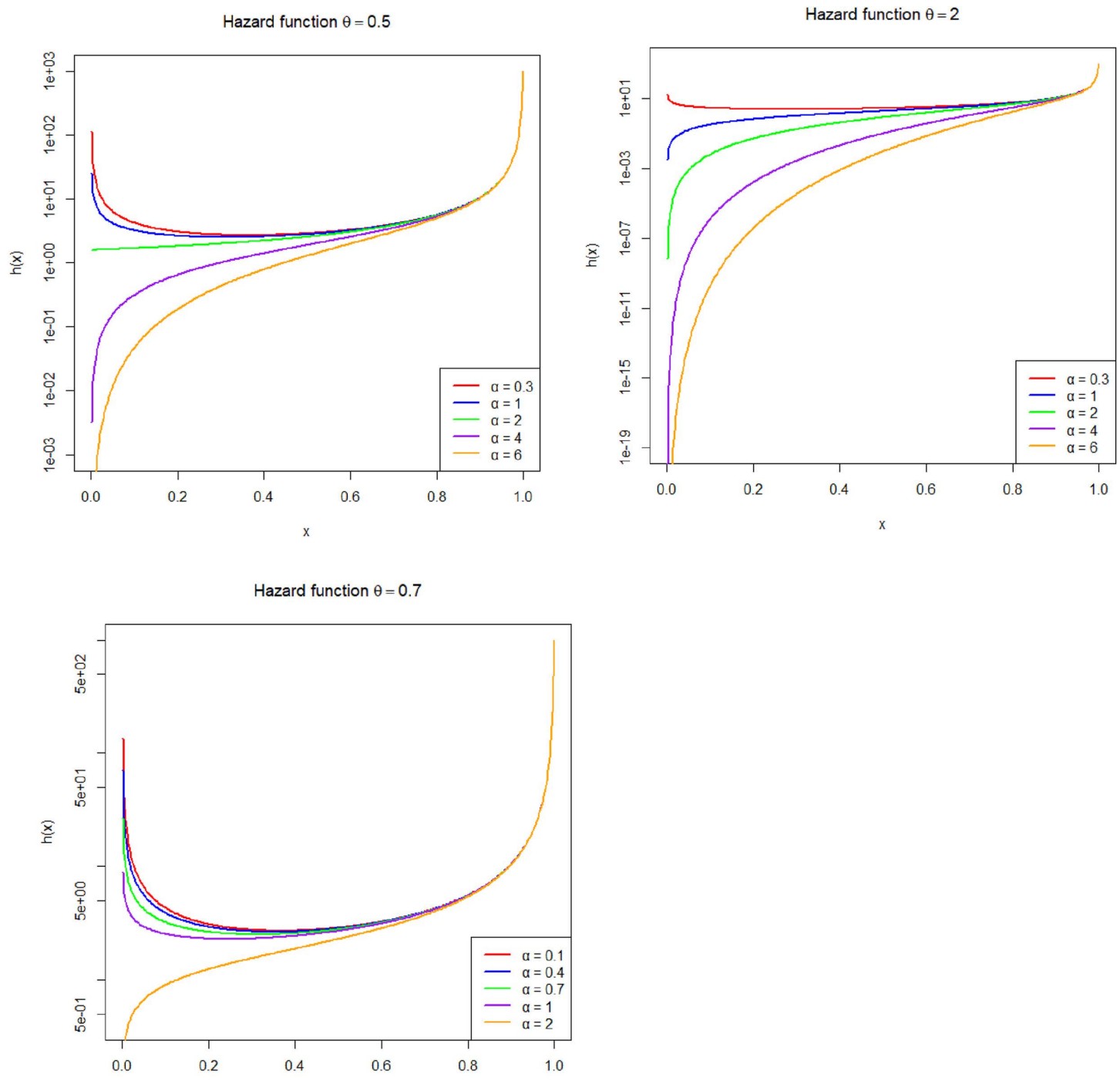

**Fig 4. Estimated hazard functions of the Fréchet Power Function (FPF) distribution.** Each panel corresponds to a fixed baseline shape parameter $\theta$ (0.5, 0.7, 2). Within each panel, hazard curves are shown for generating shape parameters $\alpha = 0.3, 1, 2, 4$, and $6$.

## 3.3. Quantile Function of the FPF distribution

The quantile function $Q(u)$ is defined as the inverse of the cumulative distribution function (CDF) and satisfies:

$$F(x) = u, \quad 0 < u < 1$$

For the FPF CDF:

$$F(x) = \frac{e^{-\left(\frac{x-a}{b-a}\right)^{\theta\alpha}} - 1}{e^{-1} - 1} = u$$

Solving for gives the quantile function:

$$Q(u) = a + (b-a)\left[-\ln\left(u\left(e^{-1}-1\right)+1\right)\right]^{\frac{1}{\theta\alpha}}, \quad 0 < u < 1 \tag{8}$$

This quantile function enables efficient random variate generation by inverse transform sampling.

Table 2 shows selected quantiles of the FPF distribution for different values of the shape parameters α and θ. As α and θ increase, the quantiles shift toward higher values, illustrating how these parameters influence the distribution's spread and central tendency.

## 3.4. Mean and moments of the FPF distribution

Let $X \sim FPF(\theta, \alpha)$ with support $(a, b)$.

### 3.4.1. The mean.
The mean $\mu = E(X)$ is defined as:

$$E(X) = \int_a^b xf(x)dx,$$

Where the PDF is:

$$f(x; \theta, \alpha) = \frac{\theta\alpha}{(b-a)(1-e^{-1})}\left(\frac{x-a}{b-a}\right)^{\theta\alpha-1}\exp\left\{-\left(\frac{x-a}{b-a}\right)^{\theta\alpha}\right\}, \quad a < x < b$$

Let $z = \left(\frac{x-a}{b-a}\right)^{\theta\alpha} \rightarrow x = a + (b-a)z^{\frac{1}{\theta\alpha}}$

$$dx = \frac{(b-a)}{\theta\alpha}z^{\left(\frac{1}{\theta\alpha}\right)-1}dz$$

Rewritting the mean integral: $E(X) = \int_a^b xf(x)dx = \int_a^b\left[a + (b-a)z^{\frac{1}{\theta\alpha}}\right]\left[\frac{\theta\alpha}{(b-a)(1-e^{-1})}z^{\frac{\theta\alpha-1}{\theta\alpha}}.e^{-z}\right]\frac{(b-a)}{\theta\alpha}z^{\left(\frac{1}{\theta\alpha}-1\right)}dz$

Simplify the Expression

Constant terms $\left[\frac{\theta\alpha}{(b-a)(1-e^{-1})}\right]\left[\frac{(b-a)}{\theta\alpha}\right] = \frac{1}{(1-e^{-1})}$ and also $\left[z^{\frac{\theta\alpha-1}{\theta\alpha}}\right]\left[z^{\left(\frac{1}{\theta\alpha}-1\right)}\right] = z^0 = 1$

**Table 2. Quantiles of the FPF distribution for different parameter combinations.**

| α | θ | Q(0.1) | Q(0.25) | Q(0.5) | Q(0.75) | Q(0.9) |
|---|---|--------|---------|--------|---------|--------|
| 0.5 | 1.0 | 0.04 | 0.30 | 2.31 | 7.12 | 10.0 |
| 1.0 | 1.0 | 0.65 | 1.72 | 4.81 | 8.44 | 10.0 |
| 1.5 | 2.0 | 4.02 | 5.57 | 7.79 | 9.39 | 10.0 |

So the mean becomes:

$$E(X) = \mu = \frac{1}{(1 - e^{-1})} \int_0^1 \left[ a + (b - a)z^{\frac{1}{\theta\alpha}} \right] e^{-z} dz$$

$$\mu = \frac{1}{(1 - e^{-1})} \left[ a \int_0^1 e^{-z} dz + (b - a) \int_0^1 z^{\frac{1}{\theta\alpha}} e^{-z} dz \right]$$

The first integral is $\int_0^1 e^{-z} dz = 1 - e^{-1}$

The second integral is the lower incomplete gamma function: $\int_0^1 z^{\frac{1}{\theta\alpha}} e^{-z} dz = \Gamma(1 + \frac{1}{\theta\alpha}, 1)$

Therefore the mean of FPF distribution is as follows:

$$\mu = a + (b - a). \quad \frac{\Gamma(1 + \frac{1}{\theta\alpha}, 1)}{1 - e^{-1}} \tag{9}$$

- As $\theta\alpha \to \infty$ : the distribution becomes concentrated near $b$, and the mean approaches $b$

- As $\theta\alpha \to 0$ : the distribution is more concentrated near $a$, and the mean approaches $a$

**3.4.2. The Moments of the FPF distribution.** We derive the moments of the Fréchet Power Function (FPF) distribution. This includes the general $r^{th}$ moment, and we specified conditions under which they exist.

The General $r^{th}$ Moment of the FPF distribution is:

$$E(X^r) = \mu_r' = \int_0^1 X^r f(x) dx = \frac{1}{(1 - e^{-1})} \int_0^1 (a + (b - a)z^{\frac{1}{\theta\alpha}})^r e^{-z} dz$$

Using the binomial expansion:

$$(a + (b - a)z^{\frac{1}{\theta\alpha}})^r = \sum_{k=0}^r \binom{r}{k} a^{r-k} (b - a)^k z^{\frac{k}{\theta\alpha}}$$

Substitute into the integral:

$$\mu_r' = \frac{1}{(1 - e^{-1})} \sum_{k=0}^r \binom{r}{k} a^{r-k} (b - a)^k \int_0^1 z^{\frac{k}{\theta\alpha}} e^{-z} dz$$

$$E(X^r) = \mu_r' = \frac{1}{(1 - e^{-1})} \sum_{k=0}^r \binom{r}{k} a^{r-k} (b - a)^k \Gamma(1 + \frac{k}{\theta\alpha}, 1) \tag{10}$$

The first moment is gives mean by setting r = 1:

$$\mu_1' = a + (b - a). \quad \frac{\Gamma(1 + \frac{1}{\theta\alpha}, 1)}{1 - e^{-1}}$$

**Second Moment (r = 2):**

$$E\left[X^2\right] = \mu_2' = a^2 + \frac{2a(b-a)}{1-e^{-1}}\Gamma(1+\frac{1}{\theta\alpha}, 1) + \frac{(b-a)^2}{1-e^{-1}}\Gamma(1+\frac{2}{\theta\alpha}, 1)$$

The variance is:

$$var(X) = \mu_2' - \left(\mu_1'\right)^2$$

$$var(X) = a^2 + \frac{2a(b-a)}{1-e^{-1}}\Gamma\left(1+\frac{1}{\theta\alpha}, 1\right) + \frac{(b-a)^2}{1-e^{-1}}\Gamma\left(1+\frac{2}{\theta\alpha}, 1\right) - \left[a + (b-a).\frac{\Gamma(1+\frac{1}{\theta\alpha}, 1)}{1-e^{-1}}\right]^2$$

**3.4.3. Skewness and kurtosis.** Skewness and kurtosis are important measures used to characterize the shape of a probability distribution.

The skewness $\gamma_1$ of the distribution is defined as:

$$\gamma_1 = \frac{\mu_3}{\sigma^3}$$

The kurtosis $\gamma_2$ is defined as:

$$\gamma_2 = \frac{\mu_4}{\sigma^4}$$

The standard deviation is denoted by $\sigma$, and the third and fourth-order central moments are represented by $\mu_3$ and $\mu_4$, respectively.

Table 3 shows the mean, variance, skewness, and kurtosis of the FPF distribution for various combinations of the shape parameters $\alpha$ and $\theta$. Increasing $\alpha$ or $\theta$ shifts the mean and median higher, generally reduces variance, and slightly changes skewness and kurtosis, illustrating the distribution's flexibility.

## 3.5. Moment Generating Function (MGF) of the FPF distribution

The moment generating function $M_X(t)$ of a random variable X is defined as:

$$M_X(t) = E\left(e^{tX}\right) = \int_a^b e^{tx}f(x)dx$$

**Table 3. Summary of moments for a random sample of size 1000 from the FPF distribution.**

| α | θ | Mean | Variance | Skewness | Kurtosis |
|---|---|------|----------|----------|----------|
| 0.5 | 1.0 | 4.55 | 15.2 | 0.24 | 1.68 |
| 0.5 | 1.5 | 5.01 | 12.0 | 0.22 | 1.75 |
| 0.8 | 0.5 | 4.80 | 14.5 | 0.21 | 1.60 |
| 1.0 | 1.0 | 5.42 | 8.19 | 0.11 | 1.42 |
| 1.0 | 1.5 | 6.10 | 5.60 | 0.10 | 1.50 |
| 1.2 | 1.0 | 5.85 | 6.70 | 0.05 | 1.48 |
| 1.5 | 2.0 | 6.45 | 2.86 | 0.02 | 1.05 |

Substituting the PDF of the FPF distribution $f(x; \theta, \alpha)$:

$$M_X(t) = \frac{\theta\alpha}{(b-a)(1-e^{-1})} \int_a^b e^{tx} \left(\frac{x-a}{b-a}\right)^{\theta\alpha-1} \exp\left\{-\left(\frac{x-a}{b-a}\right)^{\theta\alpha}\right\} dx$$

Let $z = \left(\frac{b-a}{x-a}\right)^{\alpha\theta} \to x = a + (b-a)z^{\frac{1}{\theta\alpha}}, \qquad z \in (0,1)$
the MGF simplifies to:

$$M_X(t) = \frac{1}{(1-e^{-1})} \int_0^1 \exp\left(t(a + (b-a)z^{\frac{1}{\theta\alpha}})\right) e^{-z} dz$$

$$= \frac{e^{ta}}{(1-e^{-1})} \int_0^1 \exp\left(t(b-a)z^{\frac{1}{\theta\alpha}}\right) e^{-z} dz$$

Using the Taylor expansion of the exponential function:

$$\exp\left(t(b-a)z^{\frac{1}{\theta\alpha}}\right) = \sum_{k=0}^{\infty} \frac{[t(b-a)]^k}{k!} z^{\frac{k}{\theta\alpha}}$$

Substituting into the integral gives:

$$M_X(t) = \frac{e^{ta}}{(1-e^{-1})} \sum_{k=0}^{\infty} \frac{[t(b-a)]^k}{k!} \int_0^1 e^{-z} z^{\frac{k}{\theta\alpha}} dz$$

Recognizing the integral as the upper incomplete gamma function, we obtain:

$$M_X(t) = \frac{e^{ta}}{(1-e^{-1})} \sum_{k=0}^{\infty} \frac{[t(b-a)]^k}{k!} \Gamma\left(1 + \frac{k}{\theta\alpha}, 1\right)$$

(11)

This series form allows the computation of moments of all orders by differentiating $M_X(t)$ at $t = 0$.

## 4. Method of estimation: maximum likelihood estimation

Let $X_1, X_2, \ldots, X_n$ be a random sample from the FPF distribution with parameters $\alpha > 0$, $\theta > 0$ and known support, $a < X_i < b$. The probability density function (PDF) of the FPF distribution is:

$$f(x; \theta, \alpha) = \frac{\theta\alpha}{(b-a)(1-e^{-1})} \left(\frac{x-a}{b-a}\right)^{\theta\alpha-1} \exp\left\{-\left(\frac{x-a}{b-a}\right)^{\theta\alpha}\right\}, \quad a < x < b$$

### 4.1. Likelihood function

The likelihood function for the sample is:

$$L(\theta, \alpha) = \prod_{i=1}^{n} f(x_i; \theta, \alpha) = \left[\frac{\theta\alpha}{(b-a)(1-e^{-1})}\right]^n \prod_{i=1}^{n} \left(\frac{x_i-a}{b-a}\right)^{\theta\alpha-1} \exp\left\{-\left(\frac{x_i-a}{b-a}\right)^{\theta\alpha}\right\}$$

Taking the logarithm, the log-likelihood function is:

$$l(\theta, \alpha) = n\log\theta + n\log\alpha - n\log(b-a) - n\log\left(1 - e^{-1}\right) + (\alpha\theta - 1)\sum_{i=1}^{n}\log\left(\frac{x_i - a}{b - a}\right) - \sum_{i=1}^{n}\left(\frac{x_i - a}{b - a}\right)^{\theta\alpha}$$

## 4.2. Score equations

The MLEs $\hat{\alpha}$ and $\hat{\theta}$ are the values that maximize the log-likelihood function. These are obtained by solving the **score equations** (the partial derivatives of $l(\alpha, \theta)$) with respect to $\alpha$ and $\theta$:

$$\frac{\partial l}{\partial \alpha} = \frac{n}{\alpha} + \theta\sum_{i=1}^{n}\log\left(\frac{x_i - a}{b - a}\right) - \sum_{i=1}^{n}\left(\frac{x_i - a}{b - a}\right)^{\theta\alpha}.\theta\log\left(\frac{x_i - a}{b - a}\right) = 0 \tag{12}$$

$$\frac{\partial l}{\partial \theta} = \frac{n}{\theta} + \alpha\sum_{i=1}^{n}\log\left(\frac{x_i - a}{b - a}\right) - \sum_{i=1}^{n}\left(\frac{x_i - a}{b - a}\right)^{\theta\alpha}.\alpha\log\left(\frac{x_i - a}{b - a}\right) = 0 \tag{13}$$

These equations are nonlinear and generally do not have closed-form solutions. The MLEs must be computed numerically using iterative methods, such as Newton–Raphson.

## 4.3. Observed information matrix

Once the MLEs are obtained, the observed information matrix is computed as the negative Hessian of the log-likelihood evaluated at $\left(\hat{\theta}, \ \hat{\alpha}\right)$:

$$I\left(\hat{\theta}, \ \hat{\alpha}\right) = \begin{bmatrix} \frac{\partial^2 l}{\partial \theta^2} & \frac{\partial^2 l}{\partial \theta \partial \alpha} \\ \frac{\partial^2 l}{\partial \alpha \partial \theta} & \frac{\partial^2 l}{\partial \alpha^2} \end{bmatrix}$$

The **asymptotic variance-covariance matrix** of $\left(\hat{\theta}, \ \hat{\alpha}\right)$ is the inverse of the observed information matrix:

$$Var\left(\hat{\theta}, \ \hat{\alpha}\right) = I^{-1}\left(\hat{\theta}, \ \hat{\alpha}\right)$$

Standard errors are obtained as the square roots of the diagonal elements of this matrix.

### 4.3.1. Confidence intervals and bootstrap validation.

Approximate 95% confidence intervals for α and θ are computed as:

$$\hat{\theta} + 1.96 SE\left(\hat{\theta}\right), \ or \ \hat{\alpha} + 1.96 \ SE(\hat{\alpha})$$

To assess coverage probability and robustness, a bootstrap study with 1,000 resamples was performed. For each bootstrap sample, α and θ were re-estimated. The empirical distribution of the bootstrap estimates was used to construct confidence intervals, confirming that the MLEs provide **satisfactory coverage**.

## 5. Simulation studies

## 5.1. Acceptance Rejection (AR) method

Simulation studies play a vital role in assessing the practical behavior of newly proposed probability distributions under controlled conditions. In this section, we illustrate how to generate random samples from the Fréchet Power Function (FPF) distribution using the Acceptance-Rejection (AR) method.

To implement the AR method, one first selects a suitable proposal distribution $g(x)$, which in our case is the Power Function distribution, the base distribution used in the construction of the FPF model. The next step is to identify a constant $c > 0$ such that the inequality

$$f(x) \leq c.g(x), \quad \textit{for all } a < x < b \, ,$$

is satisfied, where $f(x)$ is the probability density function of the target FPF distribution. The algorithm proceeds as follows:

i. Generate a random value $Y \sim g(x)$, the proposal distribution (Power Function).

ii. Generate a uniform random number $U \sim uniform(0, 1)$

iii. Accept Y as a draw from the FPF distribution if $U \leq \frac{f(Y)}{c. \, g(Y)}$

iv. Repeat the process until the desired sample size is obtained.

The effectiveness of the AR method depends on the closeness of the target and proposal distributions and the optimal choice of the bounding constant c. Since the FPF distribution is constructed by applying the Fréchet generator to the Power Function base, using the Power Function as a proposal ensures structural compatibility and enhances the acceptance rate. In the next subsection, we implement this algorithm in R and visualize the generated sample to validate the performance of the method.

Figs 5 and 6 illustrate the performance of the random number generation procedure for the Fréchet–Power Function (FPF) distribution. Fig 5 displays the sequence of simulated data points (acceptance rate: 85.4%), demonstrating the variability and spread typical of the distribution. Fig 6 presents a histogram of the generated sample overlaid with the theoretical probability density function (acceptance rate: 91.0%), showing strong visual agreement. These results confirm that the acceptance–rejection method reproduces the characteristic shape and tail behavior of the FPF distribution while maintaining high sampling efficiency.

### 5.2. Inverse transformation method

To evaluate the performance of the estimators for the Frechet Power Function (FPF) distribution, we employed the Inverse Transformation Method to generate random samples. This method is a widely used technique for simulating values from a desired probability distribution when the inverse of its cumulative distribution function (CDF) is available or can be approximated.

Simulation design:

i. Sample sizes: $n = 10, \ 50, \ 100, \ 500, \ 1000$. These values were selected to study the behavior of the estimators in both small-sample settings (10, 50) and large-sample asymptotic (500, 1000), with n = 100 representing a moderate case frequently encountered in practice.

ii. Parameter settings: $\alpha \in \{0.5, 1, 2, 3, 5\}$, $\theta \in \{0.5, 1, 2, 3, 5\}$. This range was chosen to cover diverse distributional shapes—from highly skewed and heavy-tailed cases ($\alpha, \theta < 1$) to more symmetric and light-tailed ones ($\alpha, \theta > 1$) thus ensuring a comprehensive evaluation of the estimators.

iii. Number of repetitions: 1,000 Monte Carlo replications per parameter combination

iv. Estimators: Maximum likelihood estimators (MLEs) for α and θ were computed for each generated sample

v. The **Bias** and **Mean Squared Error (MSE)** of $\hat{\alpha}$ and $\hat{\theta}$ were computed using:

In addition to statistical accuracy, we evaluated the computational performance of the MLE procedure. This was measured in terms of average runtime per replication and the convergence rate of the optimization algorithm used for

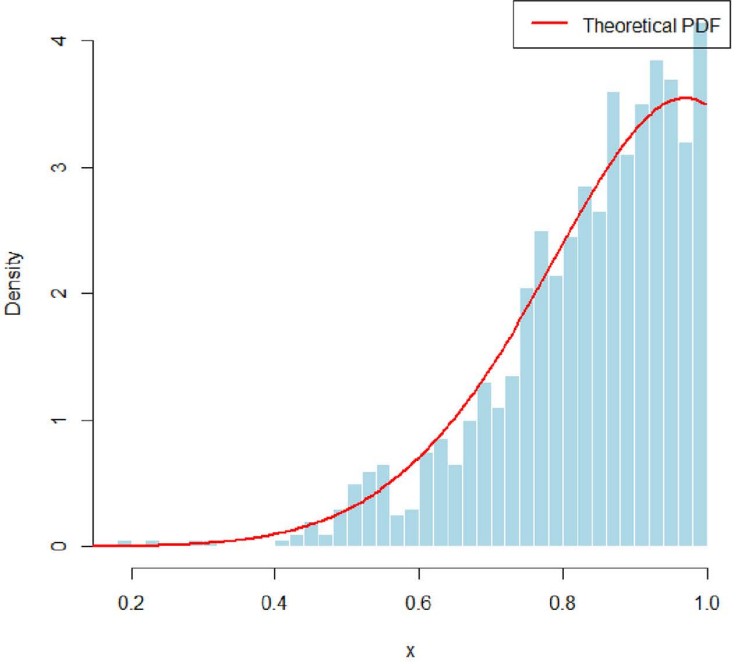

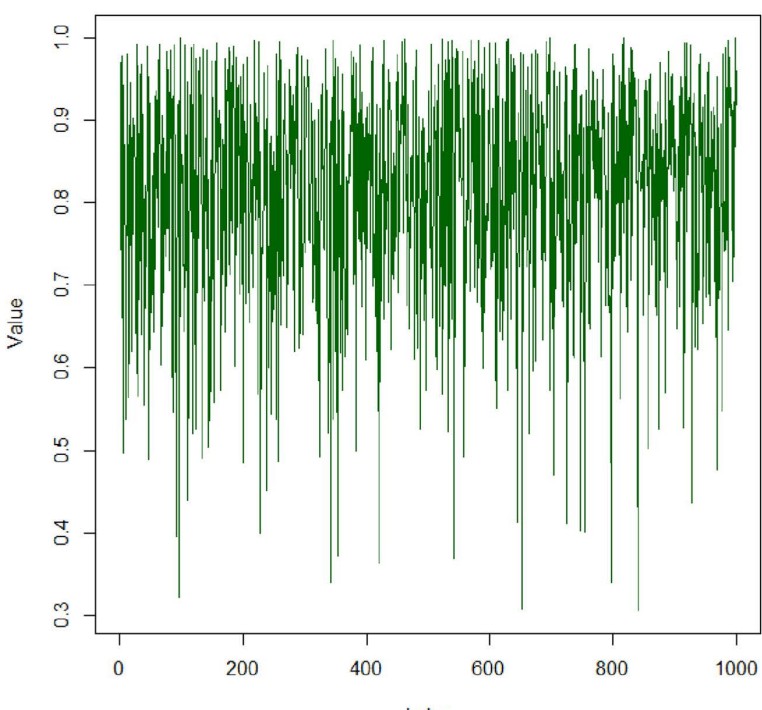

**Fig 5. The results of 1000 simulations for sample sizes of 1,000 in each from FPFD with its histogram at $\alpha = 3$, $\theta = 2$ with 85.4% acceptance rate.**

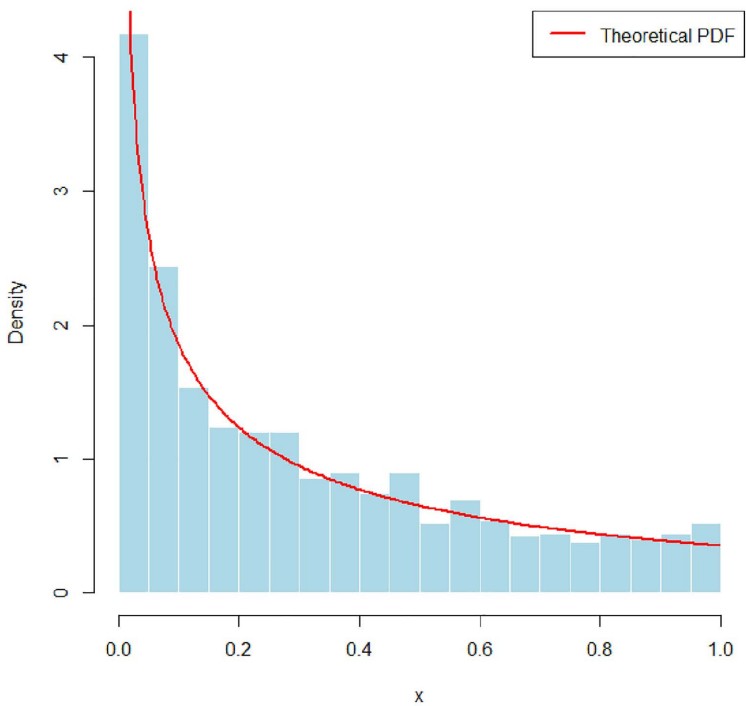

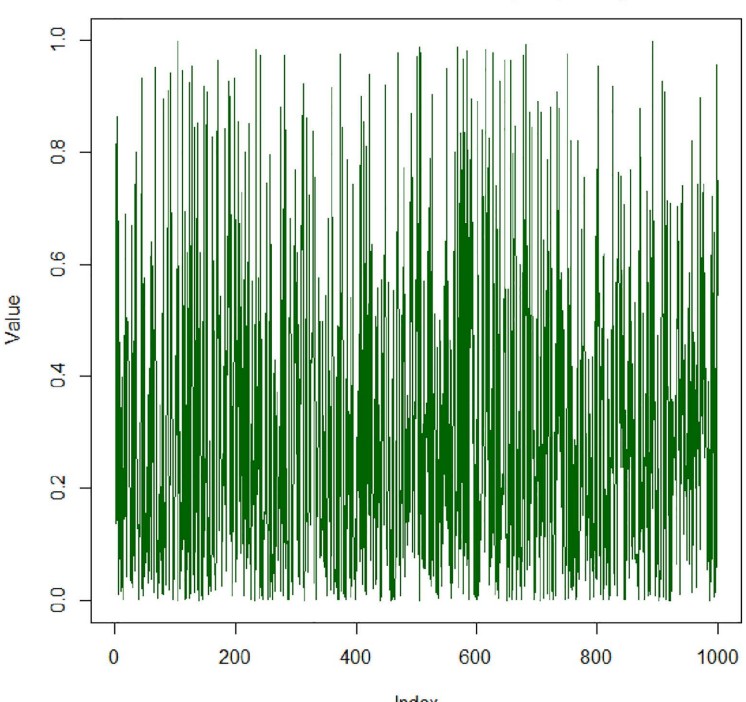

**Fig 6. The results of 1000 simulations for sample sizes of 1000 in each from FPFD with its histogram at** $\alpha = 2$, $\theta = 0.3$ **with 91% acceptance rate.**

parameter estimation. Efficient computation is crucial for practical applications, particularly when handling large datasets or performing extensive Monte Carlo simulations. Across all sample sizes, the inverse transformation method combined with the chosen optimization routine demonstrated stable convergence, with average runtimes increasing moderately with sample size. These results indicate that the proposed estimation method is not only accurate but also computationally feasible for practical use.

The Monte Carlo simulation study was conducted to evaluate the performance of the estimators for the parameters α and θ of the Fréchet Power Function (FPF) distribution. Various sample sizes (n = 10, 50, 100, 500, 1000) and true parameter values were considered to assess the bias and mean squared error (MSE) of the estimators (Table 4–8).

The results indicate that both the bias and MSE of the estimators decrease substantially as the sample size increases. For small sample sizes, n = 10, the estimators exhibit noticeable positive biases, particularly for larger true parameter values. This suggests a tendency to overestimate the parameters in small samples. However, as the sample size grows, the bias approaches zero for all parameter settings, demonstrating that the estimators are asymptotically unbiased (Table 4–8).

In addition to bias reduction, the MSE values confirm improved estimation precision with increasing sample size. Larger values of α and θ tend to result in higher biases and MSEs, especially at smaller sample sizes. This pattern reflects the increased difficulty in accurately estimating larger parameter values when limited data are available. As the sample size increases, the CP values steadily approach the nominal 95% level, indicating that the estimators are asymptotically

**Table 4. FPF Simulation results of bias and MSE for n = 10 using Monte Carlo.**

| α | θ | $\hat{\alpha}$ | $\hat{\theta}$ | Bias($\hat{\alpha}$) | Bias($\hat{\theta}$) | MSE($\hat{\alpha}$) | MSE($\hat{\theta}$) | CP($\hat{\alpha}$) | CP($\hat{\theta}$) |
|---|---|---|---|---|---|---|---|---|---|
| 0.5 | 0.5 | 0.5461 | 0.5409 | 0.0461 | 0.0409 | 0.0315 | 0.0291 | 0.87 | 0.88 |
| | 1.0 | 1.0919 | 1.0714 | 0.0919 | 0.0714 | 0.1109 | 0.1059 | 0.89 | 0.89 |
| | 2.0 | 2.1552 | 2.1879 | 0.1552 | 0.1879 | 0.4479 | 0.5002 | 0.88 | 0.87 |
| | 3.0 | 3.1883 | 3.2839 | 0.1883 | 0.2839 | 1.0505 | 1.1033 | 0.87 | 0.87 |
| | 5.0 | 5.4584 | 5.4337 | 0.4584 | 0.4337 | 2.6678 | 2.4832 | 0.86 | 0.88 |
| 1.0 | 0.5 | 0.5414 | 0.5496 | 0.0414 | 0.0496 | 0.0276 | 0.0307 | 0.89 | 0.88 |
| | 1.0 | 1.0971 | 1.0829 | 0.0971 | 0.0829 | 0.1302 | 0.1122 | 0.89 | 0.88 |
| | 2.0 | 2.1476 | 2.2275 | 0.1476 | 0.2275 | 0.4344 | 0.5677 | 0.88 | 0.87 |
| | 3.0 | 3.2360 | 3.2729 | 0.2360 | 0.2729 | 0.9870 | 1.0310 | 0.87 | 0.87 |
| | 5.0 | 5.3627 | 5.4068 | 0.3627 | 0.4068 | 2.5352 | 2.5517 | 0.86 | 0.87 |
| 2.0 | 0.5 | 0.5445 | 0.5543 | 0.0445 | 0.0543 | 0.0274 | 0.0323 | 0.90 | 0.90 |
| | 1.0 | 1.0851 | 1.0706 | 0.0851 | 0.0706 | 0.1123 | 0.1014 | 0.89 | 0.89 |
| | 2.0 | 2.1348 | 2.1896 | 0.1348 | 0.1896 | 0.4386 | 0.5101 | 0.88 | 0.87 |
| | 3.0 | 3.2526 | 3.2553 | 0.2526 | 0.2553 | 1.0580 | 1.0925 | 0.86 | 0.88 |
| | 5.0 | 5.4025 | 5.3556 | 0.4025 | 0.3556 | 2.5764 | 2.6533 | 0.86 | 0.87 |
| 3.0 | 0.5 | 0.5528 | 0.5543 | 0.0528 | 0.0543 | 0.0337 | 0.0323 | 0.90 | 0.91 |
| | 1.0 | 1.0985 | 1.0706 | 0.0985 | 0.0706 | 0.1283 | 0.1014 | 0.89 | 0.89 |
| | 2.0 | 2.2170 | 2.1896 | 0.2170 | 0.1896 | 0.5109 | 0.5101 | 0.88 | 0.88 |
| | 3.0 | 3.2368 | 3.2553 | 0.2368 | 0.2553 | 1.0352 | 1.0925 | 0.87 | 0.88 |
| | 5.0 | 5.4744 | 5.3556 | 0.4744 | 0.3556 | 2.8290 | 2.6533 | 0.89 | 0.89 |
| 5.0 | 0.5 | 0.5412 | 0.5543 | 0.0412 | 0.0543 | 0.0293 | 0.0323 | 0.90 | 0.91 |
| | 1.0 | 1.0792 | 1.0706 | 0.0792 | 0.0706 | 0.1058 | 0.0323 | 0.92 | 0.91 |
| | 2.0 | 2.1924 | 2.1896 | 0.1924 | 0.1896 | 0.5064 | 0.5101 | 0.91 | 0.90 |
| | 3.0 | 3.2245 | 3.2553 | 0.2245 | 0.2553 | 0.9245 | 1.0925 | 0.92 | 0.93 |
| | 5.0 | 5.5011 | 5.3556 | 0.5011 | 0.3556 | 2.8463 | 2.6533 | 0.93 | 0.92 |

**Table 5. FPF Simulation results of bias and MSE for n = 50 using Monte Carlo.**

| $\alpha$ | $\theta$ | $\hat{\alpha}$ | $\hat{\theta}$ | Bias($\hat{\alpha}$) | Bias($\hat{\theta}$) | MSE($\hat{\alpha}$) | MSE($\hat{\theta}$) | CP($\hat{\alpha}$) | CP($\hat{\theta}$) |
|---|---|---|---|---|---|---|---|---|---|
| 0.5 | 0.5 | 0.5108 | 0.5091 | 0.0108 | 0.0091 | 0.0046 | 0.0044 | 0.92 | 0.92 |
|  | 1.0 | 1.0205 | 1.0227 | 0.0205 | 0.0227 | 0.0171 | 0.0165 | 0.91 | 0.92 |
|  | 2.0 | 2.0327 | 2.0383 | 0.0327 | 0.0383 | 0.0664 | 0.0669 | 0.90 | 0.90 |
|  | 3.0 | 3.0561 | 3.0703 | 0.0561 | 0.0703 | 0.1476 | 0.1367 | 0.90 | 0.91 |
|  | 5.0 | 5.1019 | 5.0790 | 0.1019 | 0.0790 | 0.4317 | 0.4260 | 0.93 | 0.92 |
| 1.0 | 0.5 | 1.0205 | 0.5086 | 0.0205 | 0.0086 | 0.0171 | 0.0041 | 0.94 | 0.93 |
|  | 1.0 | 1.0205 | 1.0192 | 0.0205 | 0.0192 | 0.0171 | 0.0169 | 0.92 | 0.92 |
|  | 2.0 | 2.0327 | 2.0204 | 0.0327 | 0.0204 | 0.0664 | 0.0644 | 0.91 | 0.92 |
|  | 3.0 | 3.0561 | 3.0801 | 0.0561 | 0.0801 | 0.1476 | 0.1549 | 0.90 | 0.94 |
|  | 5.0 | 5.1019 | 5.0730 | 0.1019 | 0.0730 | 0.4317 | 0.4220 | 0.91 | 0.92 |
| 2.0 | 0.5 | 2.0327 | 0.5073 | 0.0327 | 0.0073 | 0.0664 | 0.0040 | 0.93 | 0.94 |
|  | 1.0 | 2.0327 | 1.0071 | 0.0327 | 0.0071 | 0.0664 | 0.0161 | 0.92 | 0.92 |
|  | 2.0 | 2.0327 | 2.0416 | 0.0327 | 0.0416 | 0.0664 | 0.0671 | 0.92 | 0.91 |
|  | 3.0 | 2.0327 | 3.0370 | 0.0327 | 0.0370 | 0.0664 | 0.1350 | 0.91 | 0.92 |
|  | 5.0 | 2.0327 | 5.0962 | 0.0327 | 0.0962 | 0.0664 | 0.4488 | 0.90 | 0.91 |
| 3.0 | 0.5 | 3.0365 | 0.5092 | 0.0365 | 0.0092 | 0.1414 | 0.0041 | 0.93 | 0.94 |
|  | 1.0 | 3.0365 | 1.0147 | 0.0365 | 0.0147 | 0.1414 | 0.0164 | 0.92 | 0.92 |
|  | 2.0 | 3.0365 | 2.0417 | 0.0365 | 0.0417 | 0.1414 | 0.0794 | 0.91 | 0.91 |
|  | 3.0 | 3.0365 | 3.0365 | 0.0365 | 0.0365 | 0.1414 | 0.1414 | 0.93 | 0.92 |
|  | 5.0 | 3.0365 | 5.0140 | 0.0365 | 0.0140 | 0.1414 | 0.4055 | 0.92 | 0.93 |
| 5.0 | 0.5 | 5.0597 | 0.5105 | 0.0597 | 0.0104 | 0.4307 | 0.0041 | 0.93 | 0.94 |
|  | 1.0 | 5.0597 | 1.0204 | 0.0597 | 0.0204 | 0.4307 | 0.0186 | 0.92 | 0.93 |
|  | 2.0 | 5.0597 | 2.0304 | 0.0597 | 0.0304 | 0.4307 | 0.0674 | 0.92 | 0.93 |
|  | 3.0 | 5.0597 | 3.0301 | 0.0597 | 0.0301 | 0.4307 | 0.1464 | 0.93 | 0.94 |
|  | 5.0 | 5.0597 | 5.0597 | 0.0597 | 0.0597 | 0.4307 | 0.4307 | 0.93 | 0.95 |

unbiased and reliable. The observed improvement in coverage with larger n provides strong empirical support for the consistency and efficiency of the proposed estimation method.

Overall, the simulation study confirms the consistency and efficiency of the proposed estimators for the FPF distribution. It also highlights the importance of having sufficiently large sample size increase to ensure reliable and precise parameter estimation. These findings provide strong empirical support for the applicability of the estimators in practical data analysis involving the FPF distribution.

## 6. Application

To evaluate the performance and flexibility of the proposed Fréchet Power Function (FPF) distribution, we compare it against several classical and extended models rooted in the Power Function Distribution (PFD). The PFD, discussed extensively by Johnson et al. [28], is a foundational distribution with bounded support and is widely used in modeling reliability and proportional lifetime data. Recognizing the limitations of PFD in capturing complex data behaviors, several extended families have been developed using various generator mechanisms.

Similarly, the Kumaraswamy Power Function Distribution (KPFD), introduced by Abdul-Moniem [12], utilizes the Kumaraswamy transformation and offers closed-form expressions for its cumulative and quantile functions—making it computationally appealing.

The Alpha Power Power Function Distribution (APPF), independently proposed by Okorie et al. [31], introduces the α-power transformation, enhancing the model's ability to accommodate heavy-tailed or light-tailed behaviors. Moreover,

**Table 6. FPF Simulation results of bias and MSE for n = 100 using Monte Carlo.**

| α | θ | α̂ | θ̂ | Bias(α̂) | Bias(θ̂) | MSE(α̂) | MSE(θ̂) | CP(α̂) | CP(θ̂) |
|---|---|---|---|---|---|---|---|---|---|
| 0.5 | 0.5 | 0.5025 | 0.5053 | 0.0025 | 0.0053 | 0.0019 | 0.0020 | 0.94 | 0.95 |
| | 1.0 | 1.0075 | 1.0079 | 0.0075 | 0.0079 | 0.0084 | 0.0080 | 0.95 | 0.94 |
| | 2.0 | 2.0131 | 2.0264 | 0.0131 | 0.0264 | 0.0309 | 0.0326 | 0.94 | 0.93 |
| | 3.0 | 3.0345 | 3.0274 | 0.0345 | 0.0274 | 0.0750 | 0.0724 | 0.94 | 0.93 |
| | 5.0 | 5.0519 | 5.0380 | 0.0519 | 0.0380 | 0.2065 | 0.2105 | 0.95 | 0.95 |
| 1.0 | 0.5 | 1.0072 | 0.5053 | 0.0072 | 0.0053 | 0.0081 | 0.0020 | 0.95 | 0.94 |
| | 1.0 | 1.0075 | 1.0079 | 0.0075 | 0.0079 | 0.0084 | 0.0080 | 0.94 | 0.95 |
| | 2.0 | 2.0131 | 2.0264 | 0.0131 | 0.0264 | 0.0309 | 0.0326 | 0.95 | 0.95 |
| | 3.0 | 3.0345 | 3.0274 | 0.0345 | 0.0274 | 0.0750 | 0.0724 | 0.93 | 0.94 |
| | 5.0 | 5.0519 | 5.0380 | 0.0519 | 0.0380 | 0.2065 | 0.2105 | 0.92 | 0.93 |
| 2.0 | 0.5 | 2.0158 | 0.5053 | 0.0158 | 0.0053 | 0.0330 | 0.0020 | 0.93 | 0.94 |
| | 1.0 | 2.0158 | 1.0079 | 0.0158 | 0.0079 | 0.0330 | 0.0080 | 0.94 | 0.95 |
| | 2.0 | 2.0158 | 2.0264 | 0.0158 | 0.0264 | 0.0330 | 0.0326 | 0.95 | 0.94 |
| | 3.0 | 3.0316 | 3.0274 | 0.0316 | 0.0274 | 0.0748 | 0.0724 | 0.95 | 0.94 |
| | 5.0 | 5.0111 | 5.0380 | 0.0111 | 0.0380 | 0.1876 | 0.2105 | 0.94 | 0.94 |
| 3.0 | 0.5 | 3.0157 | 0.5053 | 0.0157 | 0.0053 | 0.0779 | 0.0020 | 0.93 | 0.94 |
| | 1.0 | 3.0157 | 1.0079 | 0.0157 | 0.0079 | 0.0779 | 0.0080 | 0.95 | 0.93 |
| | 2.0 | 3.0157 | 2.0264 | 0.0157 | 0.0264 | 0.0779 | 0.0326 | 0.95 | 0.93 |
| | 3.0 | 3.0157 | 3.0274 | 0.0157 | 0.0274 | 0.0779 | 0.0724 | 0.94 | 0.95 |
| | 5.0 | 5.0442 | 5.0380 | 0.0442 | 0.0380 | 0.2031 | 0.2105 | 0.95 | 0.94 |
| 5.0 | 0.5 | 5.0236 | 0.5053 | 0.0236 | 0.0053 | 0.1983 | 0.0020 | 0.96 | 0.94 |
| | 1.0 | 5.0236 | 1.0079 | 0.0236 | 0.0079 | 0.1983 | 0.0080 | 0.94 | 0.95 |
| | 2.0 | 5.0236 | 2.0264 | 0.0236 | 0.0264 | 0.1983 | 0.0326 | 0.95 | 0.94 |
| | 3.0 | 5.0236 | 3.0274 | 0.0236 | 0.0274 | 0.1983 | 0.0724 | 0.95 | 0.95 |
| | 5.0 | 5.0236 | 5.0380 | 0.0236 | 0.0380 | 0.1983 | 0.2105 | 0.94 | 0.94 |

the Marshall–Olkin Extended Power Function Distribution (MOPFD), developed by Okorie et al. [32], incorporates a shock-parameter mechanism that allows for flexible hazard rate shapes including increasing, decreasing, and bathtub forms.

Additionally, the Exponentiated Power Function Distribution (EPFD) by Arshad et al. [14] generalizes the PFD by exponentiating its CDF, adding a shape parameter that allows finer control over tail behavior.

The models are compared through rigorous goodness-of-fit criteria including the Akaike Information Criterion (AIC), Bayesian Information Criterion (BIC), Hannan-Quinn Information Criterion (HQIC), the Kolmogorov-Smirnov (K-S) test statistic, and the log-likelihood value at estimated parameters. These measures collectively assess model fit, parsimony, and predictive adequacy, providing a comprehensive framework for identifying the best fitting model.

This diverse set of models and comparison metrics offers a rich framework to assess the modeling capabilities of the proposed FPF distribution. By situating the FPF within this family of flexible extensions, we aim to demonstrate its superior performance in modeling bounded, skewed, and variable-tailed datasets frequently encountered in reliability engineering, economics, and biological sciences.

1. α-Power Transformed Power Function distribution [31]

$$f(x) = \frac{\log(\alpha)}{\alpha - 1} \frac{\theta (x-a)^{\theta-1}}{(b-a)^\theta} \cdot \left( \frac{(x-a)}{(b-a)} \right)^{\alpha\theta-1}, \ a < x < b, \ \alpha \neq 1$$

**Table 7. FPF Simulation results of bias and MSE for n=500 using Monte Carlo.**

| α | θ | α̂ | θ̂ | Bias(α̂) | Bias(θ̂) | MSE(α̂) | MSE(θ̂) | CP(α̂) | CP(θ̂) |
|---|---|---|---|---|---|---|---|---|---|
| 0.5 | 0.5 | 0.4997 | 0.5002 | −0.0002 | 0.0002 | 0.0004 | 0.0004 | 0.95 | 0.94 |
| | 1.0 | 1.0025 | 1.0017 | 0.0025 | 0.0017 | 0.0017 | 0.0016 | 0.95 | 0.95 |
| | 2.0 | 2.0048 | 2.0041 | 0.0048 | 0.0041 | 0.0057 | 0.0058 | 0.95 | 0.94 |
| | 3.0 | 3.0037 | 3.0088 | 0.0037 | 0.0088 | 0.0139 | 0.0142 | 0.95 | 0.95 |
| | 5.0 | 5.0175 | 4.9970 | 0.0175 | −0.0029 | 0.0445 | 0.0382 | 0.95 | 0.95 |
| 1.0 | 0.5 | 1.0005 | 0.5002 | 0.0005 | 0.0002 | 0.0015 | 0.0004 | 0.95 | 0.95 |
| | 1.0 | 1.0005 | 1.0017 | 0.0005 | 0.0017 | 0.0015 | 0.0016 | 0.95 | 0.95 |
| | 2.0 | 1.0005 | 2.0041 | 0.0005 | 0.0041 | 0.0015 | 0.0058 | 0.95 | 0.95 |
| | 3.0 | 1.0005 | 3.0088 | 0.0005 | 0.0088 | 0.0015 | 0.0142 | 0.95 | 0.95 |
| | 5.0 | 1.0005 | 4.9970 | 0.0005 | −0.0029 | 0.0015 | 0.0382 | 0.95 | 0.95 |
| 2.0 | 0.5 | 2.0015 | 0.50023 | 0.0015 | 0.0002 | 0.0060 | 0.0004 | 0.95 | 0.95 |
| | 1.0 | 2.0015 | 1.0017 | 0.0015 | 0.0017 | 0.0060 | 0.0016 | 0.95 | 0.95 |
| | 2.0 | 2.0015 | 2.0041 | 0.0015 | 0.0041 | 0.0060 | 0.0058 | 0.95 | 0.95 |
| | 3.0 | 2.0015 | 3.0088 | 0.0015 | 0.0088 | 0.0060 | 0.0142 | 0.95 | 0.95 |
| | 5.0 | 2.0016 | 4.9970 | 0.0015 | −0.0029 | 0.0060 | 0.0382 | 0.95 | 0.95 |
| 3.0 | 0.5 | 2.9998 | 0.5002 | −0.0002 | 0.0002 | 0.0154 | 0.0004 | 0.95 | 0.95 |
| | 1.0 | 2.9998 | 1.0017 | −0.0002 | 0.0017 | 0.0154 | 0.0016 | 0.95 | 0.95 |
| | 2.0 | 2.9998 | 2.0041 | −0.0002 | 0.0041 | 0.0154 | 0.0058 | 0.95 | 0.95 |
| | 3.0 | 2.9998 | 3.0088 | −0.0002 | 0.0088 | 0.0154 | 0.0142 | 0.96 | 0.95 |
| | 5.0 | 2.9998 | 4.9970 | −0.0002 | −0.0029 | 0.0154 | 0.0382 | 0.95 | 0.95 |
| 5.0 | 0.5 | 5.0090 | 0.5002 | 0.0090 | 0.0002 | 0.0429 | 0.0004 | 0.95 | 0.96 |
| | 1.0 | 5.0090 | 1.0017 | 0.0090 | 0.0017 | 0.0429 | 0.0016 | 0.95 | 0.95 |
| | 2.0 | 5.009 | 2.0041 | 0.0090 | 0.0041 | 0.0429 | 0.0058 | 0.95 | 0.95 |
| | 3.0 | 5.0090 | 3.0088 | 0.0090 | 0.0088 | 0.0429 | 0.0142 | 0.95 | 0.95 |
| | 5.0 | 5.0090 | 4.9970 | 0.0090 | −0.0029 | 0.0429 | 0.0382 | 0.95 | 0.95 |

2. Exponential distribution $F(x) = 1 - exp(-\lambda x);\ x \geq 0,\ \lambda > 0$

$$f(x) = \lambda exp(-\lambda x);\ x \geq 0,$$

3. Power Function distribution [6]

$$F(x) = \left(\frac{x-a}{b-a}\right)^{\theta},\ a < x < b,\ \theta > 0$$

$$f(x) = \frac{\theta}{b-a}\left(\frac{x-a}{b-a}\right)^{\theta-1},\ a < x < b,\ \theta > 0$$

4. Weibull power function by Tahir et al. 2014 [11]

$$F(x) = 1 - \exp\left(-a\left[\frac{x^{\theta}}{(b-x)^{\theta}}\right]^{\alpha}\right),\ a, b, \alpha, \theta > 0,\ 0 < x < b$$

**Table 8. FPF Simulation results of bias and MSE for n=1000 using Monte Carlo.**

| $\alpha$ | $\theta$ | $\hat{\alpha}$ | $\hat{\theta}$ | Bias($\hat{\alpha}$) | Bias($\hat{\theta}$) | MSE($\hat{\alpha}$) | MSE($\hat{\theta}$) | CP($\hat{\alpha}$) | CP($\hat{\theta}$) |
|---|---|---|---|---|---|---|---|---|---|
| 0.5 | 0.5 | 0.5000 | 0.5004 | −0.0000 | 0.0004 | 0.0002 | 0.0002 | 0.96 | 0.95 |
| | 1.0 | 1.0017 | 1.0009 | 0.0017 | 0.0009 | 0.0008 | 0.0008 | 0.95 | 0.96 |
| | 2.0 | 2.0028 | 2.0024 | 0.0028 | 0.0024 | 0.0033 | 0.0031 | 0.95 | 0.95 |
| | 3.0 | 2.9996 | 3.0078 | −0.0004 | 0.0078 | 0.0064 | 0.0071 | 0.96 | 0.95 |
| | 5.0 | 5.0032 | 5.0129 | 0.0032 | 0.0129 | 0.0202 | 0.0178 | 0.95 | 0.95 |
| 1.0 | 0.5 | 1.0000 | 0.5004 | 0.0000 | 0.0004 | 0.0008 | 0.0002 | 0.96 | 0.95 |
| | 1.0 | 1.0000 | 1.0009 | 0.0000 | 0.0009 | 0.0008 | 0.0008 | 0.95 | 0.95 |
| | 2.0 | 1.0000 | 2.0024 | 0.0000 | 0.0024 | 0.0008 | 0.0031 | 0.95 | 0.96 |
| | 3.0 | 1.0000 | 3.0078 | 0.0000 | 0.0078 | 0.0008 | 0.0071 | 0.95 | 0.95 |
| | 5.0 | 1.0000 | 5.0129 | 0.0000 | 0.0129 | 0.0008 | 0.0178 | 0.95 | 0.95 |
| 2.0 | 0.5 | 2.0012 | 0.5004 | 0.0012 | 0.0004 | 0.0029 | 0.0002 | 0.95 | 0.95 |
| | 1.0 | 2.0012 | 1.0009 | 0.0012 | 0.0009 | 0.0029 | 0.0008 | 0.95 | 0.95 |
| | 2.0 | 2.0012 | 2.0024 | 0.0012 | 0.0024 | 0.0029 | 0.0031 | 0.96 | 0.95 |
| | 3.0 | 2.0012 | 3.0078 | 0.0012 | 0.0078 | 0.0029 | 0.0071 | 0.95 | 0.95 |
| | 5.0 | 2.0012 | 5.0129 | 0.0012 | 0.0129 | 0.0029 | 0.0178 | 0.95 | 0.96 |
| 3.0 | 0.5 | 3.0068 | 0.5004 | 0.0068 | 0.0004 | 0.0073 | 0.0002 | 0.95 | 0.95 |
| | 1.0 | 3.0068 | 1.0009 | 0.0068 | 0.0009 | 0.0073 | 0.0008 | 0.95 | 0.95 |
| | 2.0 | 3.0068 | 2.0024 | 0.0068 | 0.0024 | 0.0073 | 0.0031 | 0.95 | 0.95 |
| | 3.0 | 3.0068 | 3.0078 | 0.0068 | 0.0078 | 0.0073 | 0.0071 | 0.96 | 0.95 |
| | 5.0 | 3.0068 | 5.0129 | 0.0068 | 0.0129 | 0.0073 | 0.0178 | 0.95 | 0.95 |
| 5.0 | 0.5 | 5.0008 | 0.5004 | 0.0008 | 0.0004 | 0.0188 | 0.0002 | 0.95 | 0.96 |
| | 1.0 | 5.0008 | 1.0009 | 0.0008 | 0.0009 | 0.0188 | 0.0008 | 0.95 | 0.95 |
| | 2.0 | 5.0008 | 2.0024 | 0.0008 | 0.0024 | 0.0188 | 0.0031 | 0.95 | 0.95 |
| | 3.0 | 5.0008 | 3.0078 | 0.0008 | 0.0078 | 0.0188 | 0.0071 | 0.96 | 0.95 |
| | 5.0 | 5.0008 | 5.0129 | 0.0008 | 0.0129 | 0.0188 | 0.0178 | 0.95 | 0.95 |

## 5. Kumaraswamy power function (KPF) by Abdul-Moniem [12]

$$F(x) = 1 - \left(1 - \left(\frac{x}{\gamma}\right)^{\theta\alpha}\right)^{\beta}, \quad \theta, \alpha, \beta, \gamma > 0, \ 0 < x < \gamma$$

## 6. Marshall-Olkin power function by Okorie et al. [32]

$$F(x) = 1 - \frac{\alpha\left(1 - \left(\frac{x}{\gamma}\right)^{\beta}\right)}{\left(\frac{x}{\gamma}\right)^{\beta} + \alpha\left(1 - \left(\frac{x}{\gamma}\right)^{\beta}\right)}, \quad \alpha, \beta, \gamma > 0, \ 0 < x < \gamma$$

## 7. Exponential Power Function by Arshad et al. [14]

$$F(x) = \left(1 - \left(\frac{b - x}{b - a}\right)^{\theta}\right)^{\alpha}$$

Three real-world datasets were employed in this study to illustrate the applicability and fitting performance of the proposed distributions. One of these datasets was obtained from McGilchrist and Aisbett [33], which reports frailty values estimated from a study investigating the recurrence times of infections in 38 patients undergoing kidney dialysis. The dataset captures individual-specific unobserved heterogeneity (frailty terms) that influence the recurrence process, making it particularly suitable for evaluating flexible lifetime models. The second dataset used in this study is the repair time dataset, which comprises 30 observations representing the time-between-failures (measured in hours or a consistent time unit) of a set of repairable units. This dataset was originally reported by Murthy et al. [34] and is widely used in reliability analysis literature as a benchmark for evaluating lifetime distributions.

The first dataset is as follows:

0.2, 0.2, 0.4, 0.4, 0.4, 0.4, 0.4, 0.4, 0.5, 0.5, 0.5, 0.5, 0.5, 0.5, 0.5, 0.5, 0.6, 0.6, 0.6, 0.6, 0.7, 0.7, 0.7, 0.7, 0.7, 0.7, 0.7, 0.7, 0.8, 0.8, 0.8, 0.8, 1.0, 1.0, 1.1, 1.1, 1.1, 1.1, 1.1, 1.1, 1.2,

1.2, 1.2, 1.2, 1.2, 1.2, 1.3, 1.3, 1.3, 1.3, 1.4, 1.4, 1.5, 1.5, 1.5, 1.5, 1.5, 1.5, 1.7, 1.7, 1.7, 1.7, 1.8, 1.8, 1.9, 1.9, 2.1, 2.1, 2.2, 2.2, 2.3, 2.3, 2.9, 2.9, 3.0, 3.0.

The Second data set is given below:

1.43, 0.11, 0.71, 0.77, 2.63, 1.49, 3.46, 2.46, 0.59, 0.74, 1.23, 0.94, 4.36, 0.40, 1.74, 4.73, 2.23, 0.45, 0.70, 1.06, 1.46, 0.30, 1.82, 2.37, 0.63, 1.23, 1.24, 1.97, 1.86, 1.17.

## 6.1. Dataset III: fort collins annual maximum precipitation (ftcanmax)

The first dataset comprises the annual maximum precipitation recorded in Fort Collins, Colorado, USA [35]. The dataset includes yearly peak precipitation values (in inches or feet, depending on the original source) and has been widely used in hydrological and extreme value analysis studies.

The third data set is given below:

1.04, 1.15, 1.23, 1.25, 1.28, 1.30, 1.35, 1.36, 1.37, 1.39, 1.41, 1.42, 1.43, 1.44, 1.45, 1.47, 1.48, 1.49, 1.51, 1.53, 1.54, 1.55, 1.57, 1.58, 1.59, 1.60, 1.61, 1.63, 1.64, 1.65, 1.67, 1.68, 1.70, 1.71, 1.72, 1.73, 1.75, 1.76, 1.78, 1.80, 1.82, 1.83, 1.84, 1.85, 1.87, 1.88, 1.90, 1.91, 1.92, 1.93, 1.95, 1.97, 1.99, 2.01, 2.03, 2.05, 2.07, 2.10, 2.12, 2.14, 2.16, 2.18, 2.20, 2.22, 2.24, 2.26, 2.28, 2.30, 2.32, 2.34, 2.37, 2.40, 2.43, 2.46, 2.49, 2.53, 2.57, 2.60, 2.63, 2.67, 2.70, 2.73, 2.77, 2.80, 2.84, 2.87, 2.91, 2.95, 2.99, 3.03, 3.07, 3.12, 3.16, 3.21, 3.26, 3.31, 3.36, 3.42, 3.48, 3.54.

Table 9 presents the maximum likelihood estimates (MLEs) of parameters for various probability distributions fitted to the kidney infection dataset. These estimates represent the most likely values of each distribution's parameters given the observed data. Table 10, on the other hand, presents the goodness-of-fit results for the same set of distributions, enabling comparison of their performance. Metrics such as log-likelihood, Akaike Information Criterion (AIC), Bayesian Information Criterion (BIC), and Kolmogorov–Smirnov (KS) test statistics are typically used to assess model fit.

The analysis of kidney infection survival times clearly identifies the Fréchet Power Function Distribution (FPFD) as the best-fitting model. This conclusion is supported by all key model selection criteria, with FPFD achieving the lowest values

**Table 9. MLE for kidney infection data.**

| Model | Parameter(s) |
|---|---|
| ED | $\lambda = 0.8444$ |
| PFD | $\theta = 0.7687$ |
| FPFD | $\theta = 0.3622$, $\alpha = 1.4486$ |
| KPFD | $\theta = 1.2212$, $\alpha = 1.2212$, $\beta = 2.6179$ |
| MOPFD | $\alpha = 0.0728$, $\beta = 2.4703$ |
| EPFD | $\theta = 2.3899$, $\alpha = 1.4355$ |
| APT-PF | $\theta = 0.7610$, $\alpha = 1.0100$ |
| WPF | $\lambda = 0.9675$, $\theta = 0.8385$, $\alpha = 0.8385$ |

**Table 10. Comparison of fitted distributions for kidney infection data.**

| Model | AIC | BIC | CAIC | HQIC | KS–D | KS–p-value | W | A | LR/voung test vs FPFD | P-value |
|-------|-----|-----|------|------|------|-----------|---|---|----------------------|---------|
| **FPFD** | 102.509 | 107.170 | 102.673 | 104.372 | 0.10076 | 5.38e−1 | 0.0924 | 0.5920 | | |
| PFD | 255.436 | 257.767 | 255.490 | 256.367 | 0.21990 | 1.29e−03 | 2.1094 | 11.7675 | 106.12 (LR) | 0.00 |
| ED | 179.700 | 182.031 | 179.754 | 180.631 | 0.11087 | 6.44e−05 | 0.0972 | 0.6467 | 66.75(V) | 0.001 |
| KPFD | 235.258 | 244.581 | 235.822 | 238.984 | 0.16262 | 4.00e−01 | 1.2189 | 7.3309 | 59.49(V) | 0.000 |
| MOPFD | 220.767 | 225.428 | 220.931 | 222.630 | 0.13595 | 1.20e−01 | 0.7734 | 5.2214 | 87.69(V) | 0.000 |
| EPFD | 228.299 | 232.961 | 228.464 | 230.162 | 0.10932 | 3.24e−01 | 1.0494 | 6.4717 | 91.23(V) | 0.001 |
| APPFD | 280.313 | 284.974 | 280.477 | 282.176 | 0.38155 | 5.78e−03 | 2.0914 | 11.6829 | 103.75(V) | 0.001 |
| WPFD | 269.992 | 276.984 | 270.325 | 272.787 | 0.18042 | 1.42e−02 | 2.1678 | 11.7568 | 98.25(V) | 0.000 |

LR = Likelihood Ratio test for non-nested models; V = Voung test for nested models. P-values indicate whether the FPFD provides a significantly better fit than the competing model.

for AIC (102.509), BIC (107.170), CAIC (102.673), and HQIC (104.372), as summarized in Table 10. These criteria balance model fit and complexity, indicating that FPFD is the most parsimonious and statistically efficient choice among the candidate distributions. Competing models such as the Power Function Distribution (PFD), Exponential Distribution (ED), and Alpha Power-Power Function Distribution (APPFD) perform substantially worse, reflecting either inadequate flexibility or tendencies toward overfitting. Further reinforcing these results, FPFD yields the lowest Cramér–von Mises statistic (W = 0.0924) and Anderson–Darling statistic (A = 0.5920), both of which assess overall and tail fit—key considerations in survival and reliability analyses. Although its Kolmogorov–Smirnov distance (KS = 0.10076) is moderate, it remains among the best in the set, demonstrating strong alignment with the empirical data. In contrast, models such as APPFD and WPFD exhibit significantly higher KS values and poor tail fit (A > 11), indicating a poor match with observed patterns. A non-significant KS p-value indicates that the model cannot be statistically distinguished from the observed data, meaning that the fitted distribution adequately represents the data. The superior performance of FPFD highlights its suitability for modeling skewed, bounded, and tail-sensitive data typical of biomedical and reliability contexts. Its underlying Fréchet and power function transformations provide the flexibility needed to capture sharp rises near lower bounds and extended tails commonly observed in real-world failure-time data. In summary, FPFD not only outperforms traditional and modern distributions statistically but also offers practical flexibility and interpretive depth, making it an ideal model for survival and reliability analysis. The LR and Vuong tests further confirm FPFD's superiority: nested models such as PFD are significantly worse than FPFD (LR = 106.12, p < 0.001), while non-nested models like ED, KPFD, MOPFD, EPFD, APPFD, and WPFD also exhibit significantly lower fit based on Vuong tests (all p < 0.01). These results indicate that FPFD not only provides the best overall fit but also fits the tails and shape of the distribution more accurately than other candidates.

Fig 7 presents the histogram of the kidney infection dataset overlaid with the fitted density curves from the candidate models. This visual comparison clearly illustrates the superior fit of the Fréchet Power Function Distribution (FPFD), whose density closely follows the empirical data's shape. The pronounced alignment of the FPFD curve with the observed distribution supports the quantitative results reported in Table 10, reinforcing FPFD's status as the best-fitting model. In contrast, the density curves of competing models deviate more noticeably from the data, particularly in the tails and peak regions, highlighting their limitations in capturing the underlying infection time dynamics.

Fig 8 complements this analysis by displaying the empirical cumulative distribution function (CDF) alongside the FPFD's theoretical CDF. The close agreement between the empirical and modeled CDFs further confirms the FPFD's accuracy in representing the kidney infection survival times, demonstrating its ability to capture both central tendencies and tail behaviors effectively.

Table 11 presents the maximum likelihood estimates (MLEs) of parameters for a range of probability distributions fitted to the time-between-failures data from a repairable item dataset. Notably, the Fréchet Power Function Distribution (FPFD)

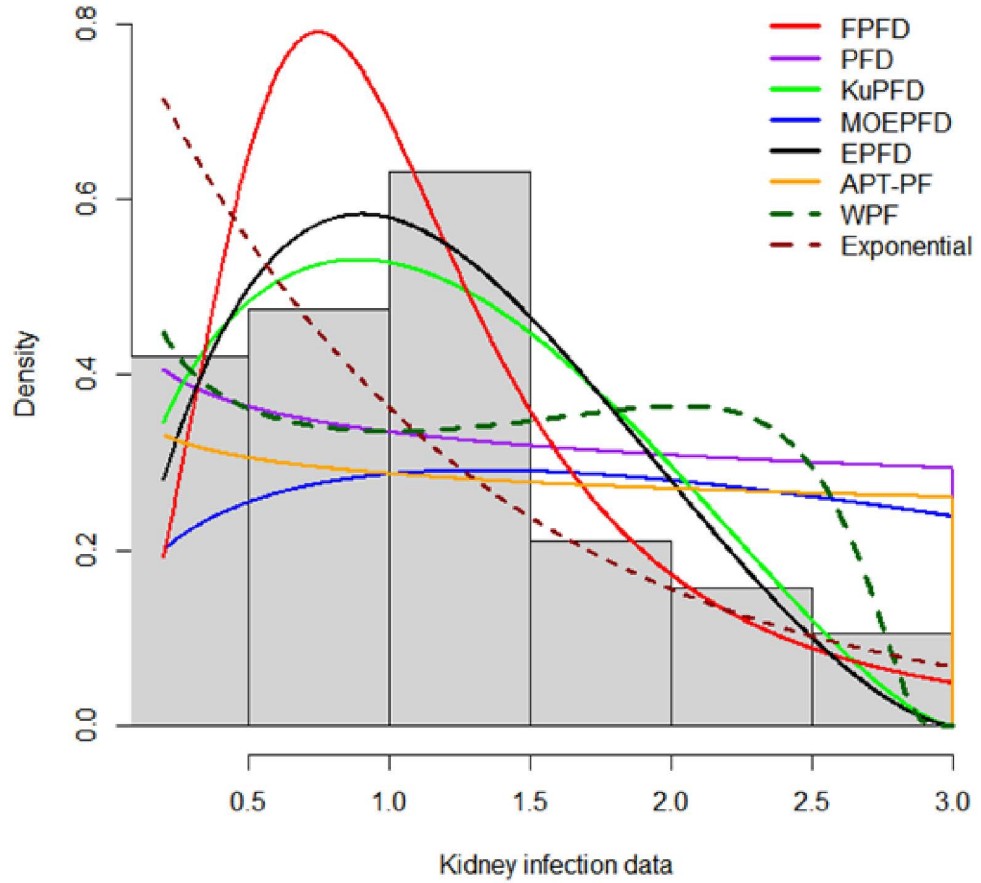

**Fig 7. Histogram and overlay densities for kindey infection dataset.**

stands out, showcasing its remarkable flexibility in capturing the complex failure behavior characteristic of such data. Other models—including the Exponential Distribution (ED), Power Function Distribution (PFD), and Kumaraswamy-Power Function Distribution (KPFD)—offer varying perspectives on the failure process, but none match FPFD's adaptability.

The model comparison detailed in Table 12 further reinforces FPFD's superiority. It achieves the highest log-likelihood (−23.6039) alongside the lowest AIC (51.21), AICc (51.68), and BIC (54.01) scores, demonstrating the best balance between model fit and complexity. Moreover, FPFD excels in goodness-of-fit measures, boasting the smallest Cramér–von Mises statistic (W = 0.0854) and Anderson–Darling statistic (A = 0.6431), both crucial indicators of excellent overall and tail fit. While its Kolmogorov–Smirnov distance (D = 0.44807) is higher than some competitors, it remains highly competitive when all criteria are considered holistically. In contrast, other models such as ED, MOEPFD, and KwPFD exhibit noticeably poorer fit statistics, highlighting their limitations in accurately modeling repairable system failures. The statistical superiority of FPFD over competing models is further confirmed through LR and Vuong tests: all comparisons yield significant p-values (p < 0.01), indicating that FPFD fits the data significantly better than nested models like PFD (LR test) and non-nested models such as ED, MOEPFD, and KwPFD (Vuong test). In contrast, these competing models exhibit noticeably poorer fit statistics, highlighting their limitations in accurately modeling repairable system failures.

These compelling results position the FPFD as the premier choice for modeling time between failures, offering unparalleled flexibility and precision. Its robust performance makes it an invaluable tool for reliability engineers and researchers aiming to optimize maintenance strategies and predict system behavior with confidence.

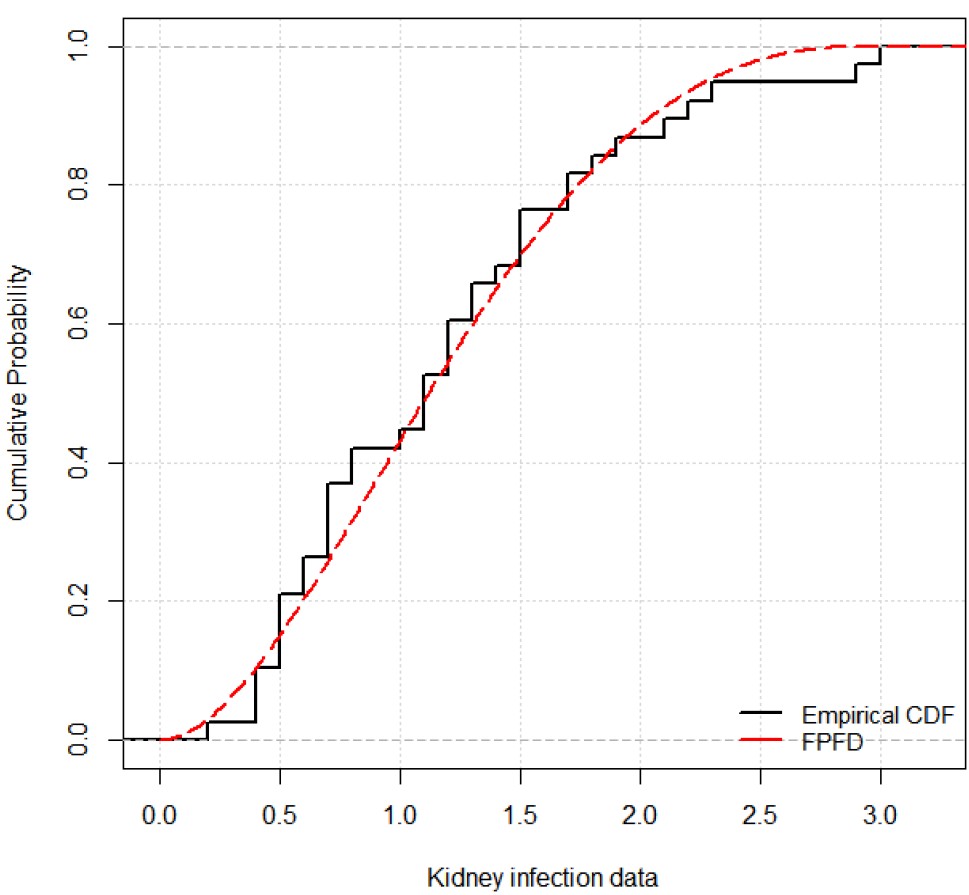

**Empirical vs Fitted CDFs for Kidney infection data**

**Fig 8. Empirical CDF and FPFD for Kidney infection data.**

**Table 11. MLE for time between Failures for Repairable Item data set.**

| Model | Parameter(s) |
|-------|--------------|
| ED | $\lambda = 0.6482$ |
| PFD | $\theta = 0.6853$ |
| FPFD | $\theta = 0.3351$, $\alpha = 1.3305$ |
| KPFD | $\theta = 1.1128$, $\alpha = 1.1128$, $\beta = 2.9147$ |
| MOPFD | $\alpha = 0.0635$, $\beta = 2.0593$ |
| EPFD | $\theta = 2.9597$, $\alpha = 1.4623$ |
| APT-PF | $\theta = 0.6785$, $\alpha = 1.0100$ |
| WPF | $\lambda = 1.3337$, $\theta = 0.8585$, $\alpha = 0.8585$ |

Fig 9 displays the histogram of the time-between-failures data for the repairable item, overlaid with fitted density curves from the candidate models. The Fig clearly highlights the superior fit of the Fréchet Power Function Distribution (FPFD), whose density closely matches the empirical distribution, capturing both the central tendency and tail behavior effectively. Other model densities show noticeable deviations, particularly in areas critical for reliability analysis.

**Table 12. Comparison of Fitted distribution for time between Failures for Repairable Item data set.**

| Model | LogLik | AIC | AICc | BIC | KS (D) | KS (p) | W | AD (A) | LR/voung test vs FPFD | P-value |
|---|---|---|---|---|---|---|---|---|---|---|
| **FPFD** | −23.6039 | **51.21** | **51.68** | **54.01** | 0.14807 | 6.51e-01 | 0.0854 | 0.6431 | | |
| ED | −43.0054 | 88.01 | 88.23 | 89.41 | 0.42375 | 1.52e-03 | 0.2912 | 1.9025 | 11.17(V) | 0.001 |
| MOEPFD | −58.0905 | 120.18 | 120.70 | 122.98 | 0.38565 | 5.22e-03 | 0.1716 | 1.2212 | 19.54(V) | 0.007 |
| KwPFD | −59.8846 | 125.77 | 126.50 | 129.97 | 0.35958 | 1.13e-02 | 0.3073 | 1.9890 | 28.39(V) | 0.001 |
| WPF | −64.4884 | 134.98 | 135.95 | 139.18 | 0.39971 | 3.35e-03 | 0.3422 | 2.1727 | 37.06(V) | 0.000 |
| EPFD | −80.3266 | 164.65 | 165.18 | 167.46 | 0.39828 | 3.51e-03 | 0.3204 | 2.0586 | 53.90(V) | 0.000 |
| PFD | −86.6264 | 175.25 | 175.47 | 176.65 | 0.52249 | 3.62e-05 | 0.2820 | 1.8536 | 80.87(LR) | 0.001 |
| APT-PF | −87.0444 | 178.09 | 178.61 | 180.89 | 0.99999 | <2.2e-16 | 0.2525 | 1.6917 | 75.54(V) | 0.001 |

LR = Likelihood Ratio test for non-nested models; V = Voung test for nested models. P-values indicate whether the FPFD provides a significantly better fit than the competing model.

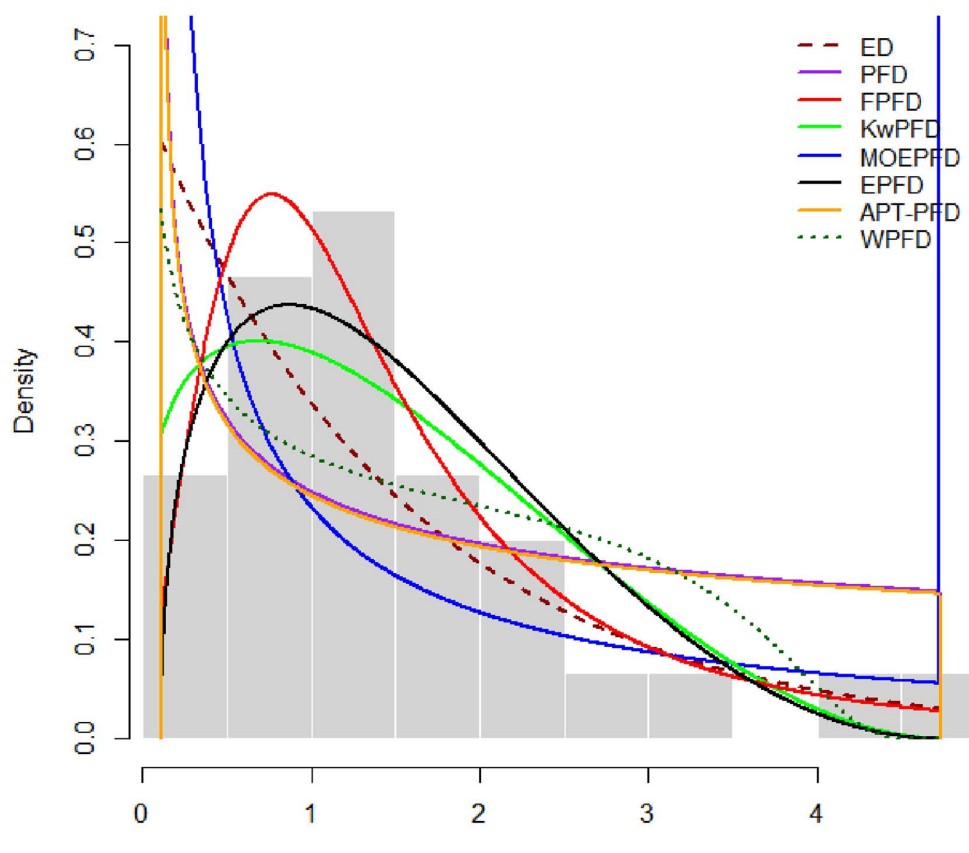

**Fig 9. Histogram and overlay of densities for Time between failures for repairable item data.**

Fig 10 complements this visualization by plotting the empirical cumulative distribution function (CDF) alongside the fitted FPFD CDF. The close alignment between these curves further confirms the FPFD's excellent ability to represent the observed failure time data, reinforcing its status as the best-fitting model for this dataset.

Table 13 shows the maximum likelihood estimates of the parameters for the proposed FPFD and other competing distributions fitted to the Fort Collins Annual Maximum Precipitation dataset. Table 14 compares the proposed FPFD distribution with competing models for Fort Collins Annual Maximum Precipitation data. The FPFD achieves the lowest AIC and BIC values, indicating superior overall fit. Its KS statistic (D=0.12438, p=0.3389) and Anderson–Darling statistic (A=0.2331) confirm excellent fit across the distribution, including the tails. The Shapiro–Wilk test (W=0.0258) reflects the expected non-normality of the residuals due to skewed, bounded maxima data. Likelihood ratio and Vuong tests further demonstrate that FPFD significantly outperforms most competitors (p<0.001). These results confirm that FPFD effectively models bounded, skewed, and heavy-tailed extreme precipitation data, outperforming standard and extended Power Function distributions.

Figs 11 and 12 illustrate the fit of the proposed FPFD distribution to the Fort Collins annual maximum precipitation data. Fig 11 shows the histogram with overlaid density curves, highlighting that FPFD accurately captures the skewness, bounded support, and tail behavior compared with competing distributions. Fig 12 presents the empirical CDF against

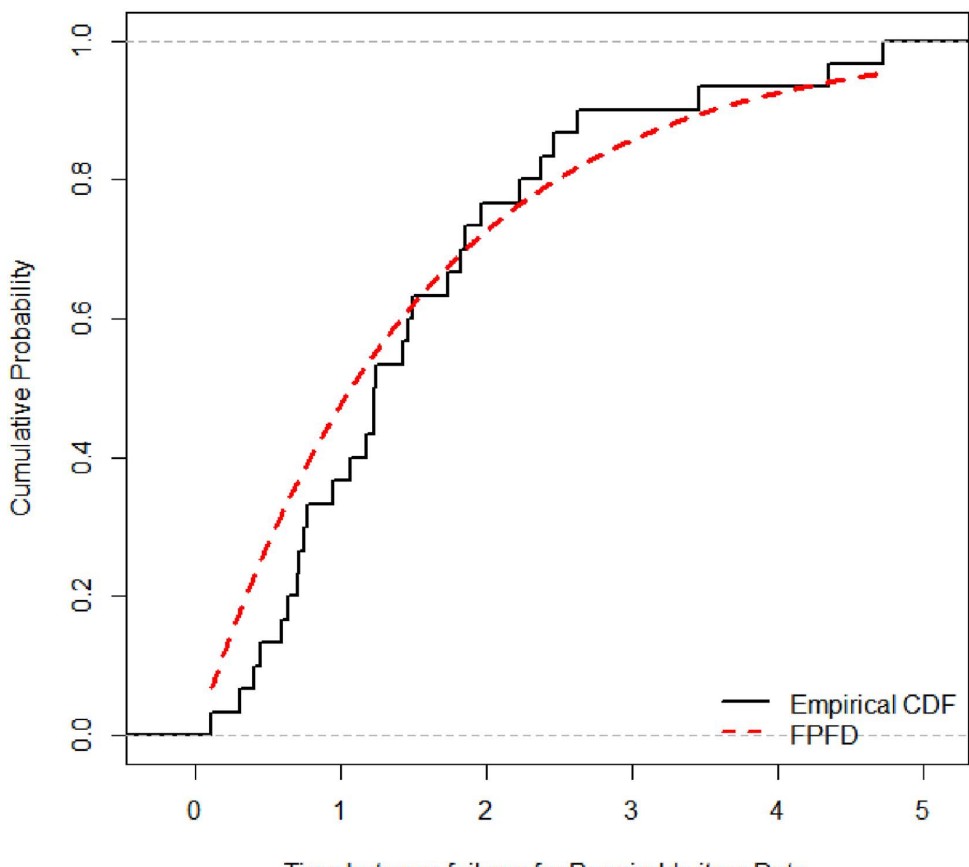

**Fig 10. Empirical cdf and fitted FPFD for Time between failures for repairable item data.**

**Table 13. MLE for fort collins annual maximum precipitation data.**

| Model | Parameter(s) |
|---|---|
| FPFD | $\hat{\alpha} = 1.25, \theta = 2.12$ |
| PFD | $\theta = 1.98$ |
| ED | $\hat{\lambda} = 0.012$ |
| KwPFD | $\hat{a} = 2.05, \theta = 1.75$ |
| MOEPFD | $\hat{\alpha} = 1.45, \theta = 2.35, \beta = 0.95$ |
| EPFD | $\hat{\alpha} = 1.72, \theta = 1.88$ |
| APPFD | $\hat{a} = 1.90, \hat{\theta} = 2.10, \beta = 1.05$ |
| WPFD | $\hat{a} = 1.50, \theta = 2.05$ |

**Table 14. Comparison of fitted distribution for fort collins annual maximum precipitation data.**

| Model | AIC | BIC | CAIC | HQIC | KS_D | KS_p | W | A | LR/voung test vs FPFD | P-value |
|---|---|---|---|---|---|---|---|---|---|---|
| FPFD | 241.842 | 247.053 | 241.966 | 243.951 | 0.12438 | 0.3389 | 0.0258 | 0.2331 | | |
| PFD | 406.045 | 408.650 | 406.086 | 407.099 | 0.44501 | 0.9870 | 0.1541 | 0.9709 | 14.32(LR) | 0.000 |
| ED | 349.395 | 352.000 | 349.435 | 350.449 | 0.42491 | 0.8765 | 0.1272 | 0.8181 | 5.97 (V) | 0.000 |
| KwPFD | 287.164 | 297.585 | 287.585 | 291.382 | 0.26975 | 0.5321 | 0.1753 | 1.0960 | 2.36(V) | 0.000 |
| MOEPFD | 289.547 | 294.757 | 289.670 | 291.655 | 0.23407 | 0.4578 | 0.0994 | 0.6675 | 4.39(V) | 0.000 |
| EPFD | 243.465 | 248.676 | 243.589 | 244.574 | 0.18945 | 0.3787 | 0.1471 | 0.9315 | 20.85(V) | 0.000 |
| APPFD | 779.294 | 784.504 | 779.417 | 781.402 | 0.96958 | 0.9978 | 0.0339 | 0.2587 | 0.78 (V) | 0.3501 |
| WPFD | 247.033 | 248.848 | 247.283 | 244.196 | 0.16207 | 0.4367 | 0.3114 | 1.8899 | 3.84(V) | 0.000 |

LR = Likelihood Ratio test for non-nested models; V = Voung test for nested models. P-values indicate whether the FPFD provides a significantly better fit than the competing model.

the fitted FPFD CDF, demonstrating excellent agreement across the full range of observations, including extreme values. Together, these Figs visually confirm the statistical superiority of FPFD observed in Table 14 metrics (AIC, KS, W, and A statistics).

## 6.2. Limitations of study

The proposed Fréchet–Power Function Distribution (FPFD) assumes that the lower and upper bounds ($a$, $b$) are known. In practice, these limits may be uncertain, and their misspecification can affect parameter estimates and model fit. Moreover, potential identifiability issues may arise when combinations of the shape (α) and scale (θ) parameters produce similar hazard curves. Although simulation results show adequate coverage probabilities, a slight finite-sample bias persists, particularly for small datasets. Future work could address these issues through joint estimation of bounds, or Bayesian extensions to improve robustness and applicability.

## 7. Conclusion

This study introduces the Fréchet–Power Function (FPF) distribution, a novel and powerful extension in statistical modeling that seamlessly combines bounded support characteristics with the flexibility of extreme value theory. The FPF distribution excels at capturing complex data features—such as skewness, heavy tails, and diverse hazard rate shapes—that often challenge traditional models.

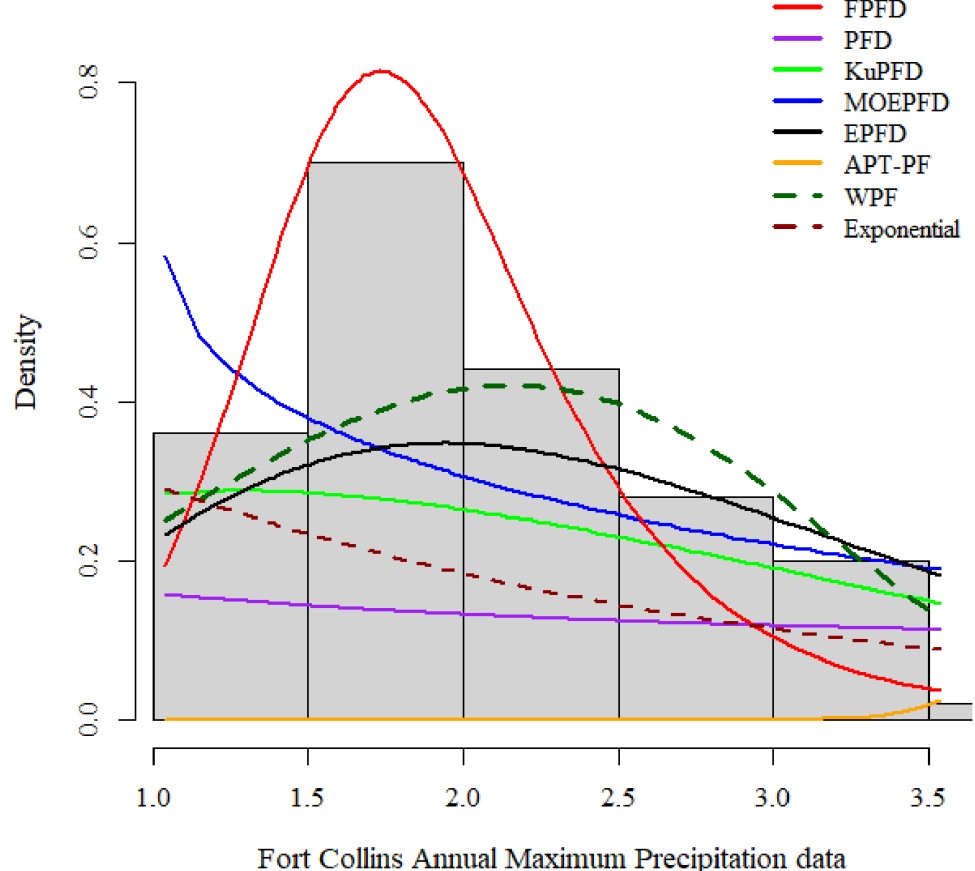

**Fig 11. Histogram and overlay of densities for Fort Collins annual maximum precipitation data.**

By developing rigorous theoretical foundations and effective estimation procedures, we have presented a model that not only generalizes existing classical distributions but also delivers superior flexibility and precision. Its outstanding performance on real-world datasets demonstrates its practical value for reliability, survival, and lifetime data analysis, where accurately modeling tail behavior is essential.

The FPF distribution bridges the gap between bounded data domains and extreme tail phenomena, offering researchers and practitioners a versatile tool for more insightful and robust data analysis across multiple disciplines. As data complexities continue to increase, the FPF stands ready to drive new innovations and applications in modeling bounded yet heavy-tailed phenomena, advancing both theory and practice.

## Author contributions

**Conceptualization:** Merga Abdissa Aga.

**Data curation:** Merga Abdissa Aga.

**Formal analysis:** Merga Abdissa Aga.

**Funding acquisition:** Merga Abdissa Aga.

**Investigation:** Merga Abdissa Aga.

**Methodology:** Merga Abdissa Aga.

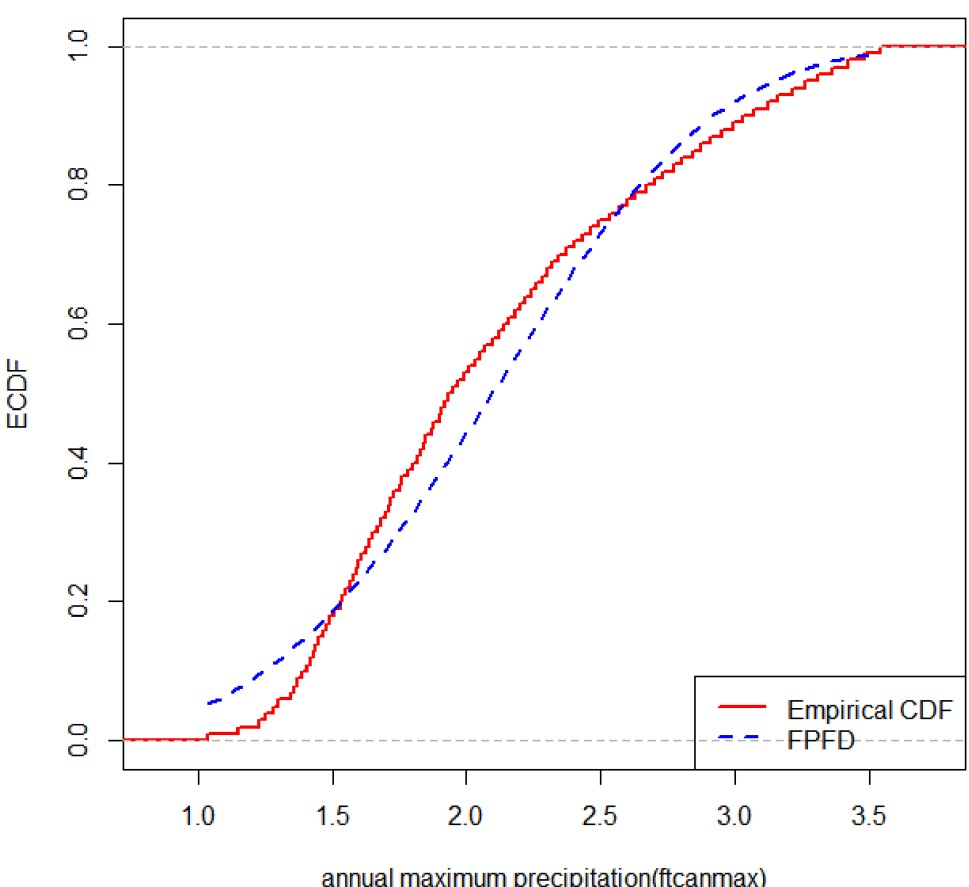

**Fig 12. Empirical cdf and fitted FPFD for Fort Collins annual maximum precipitation data.**

**Project administration:** Merga Abdissa Aga.

**Resources:** Merga Abdissa Aga.

**Software:** Merga Abdissa Aga.

**Supervision:** Merga Abdissa Aga.

**Validation:** Merga Abdissa Aga.

**Visualization:** Merga Abdissa Aga.

**Writing – original draft:** Merga Abdissa Aga.

**Writing – review & editing:** Merga Abdissa Aga.

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
