## [Decision Letter · Decision Letter 0]

10 Sep 2025

Dear Dr. Aga,

Thank you for submitting your manuscript to PLOS ONE. After careful consideration, we feel that it has merit but does not fully meet PLOS ONE’s publication criteria as it currently stands. Therefore, we invite you to submit a revised version of the manuscript that addresses the points raised during the review process.

We look forward to receiving your revised manuscript.

Kind regards,

Fucai Lin, Ph.D.

Academic Editor

PLOS ONE

Journal Requirements:

Reviewers' comments:

Reviewer's Responses to Questions

**Comments to the Author**

1. Is the manuscript technically sound, and do the data support the conclusions?

Reviewer #1: Yes

Reviewer #2: Partly

2. Has the statistical analysis been performed appropriately and rigorously?

Reviewer #1: Yes

Reviewer #2: Yes

3. Have the authors made all data underlying the findings in their manuscript fully available?

Reviewer #1: Yes

Reviewer #2: Yes

4. Is the manuscript presented in an intelligible fashion and written in standard English?

Reviewer #1: Yes

Reviewer #2: Yes

Reviewer #1: Summary

This manuscript introduces the Fréchet–Power Function (FPF) distribution, a novel bounded-support model with heavy-tailed flexibility, designed to overcome limitations in existing bounded lifetime models. The authors provide theoretical development, statistical properties, estimation procedures, simulation studies, and real-world applications. The work is well-motivated and mathematically sound, and it addresses a genuine gap in the literature.

However, the manuscript would benefit from significant revisions to improve clarity, strengthen empirical validation, and better position the model within the broader statistical literature. My detailed comments are organized below.

Major Points

Positioning in Literature

The introduction cites relevant models but lacks a clear structural framework distinguishing bounded-only, heavy-tailed-only, and hybrid models.

A comparative summary table of existing models versus the FPF in terms of support, tail behavior, and hazard flexibility would help highlight novelty.

Methodological Clarity

Some derivations (e.g., Theorem 1) contain redundant steps and inconsistent notation (αθ vs. θα).

Proof readability would improve if derivations were streamlined and some moved to supplementary material.

Parameter Identifiability

The hazard function section does not address whether different parameter combinations can produce indistinguishable hazard shapes, which could impact interpretability.

Simulation Study

Bias and MSE results are thorough, but coverage probabilities, computational performance, and acceptance–rejection efficiency metrics are missing.

Including these would strengthen the empirical evaluation of the estimation methods.

Applications and Model Comparison

The comparison methodology (criteria, estimation method for competitors) is not fully detailed.

The results would be more convincing with likelihood ratio tests, Vuong tests, and multi-criteria fit assessment (AIC, BIC, KS, Anderson–Darling).

Assumption of Known Bounds

The model assumes (a, b) are known, but in real data this may not be the case.

A sensitivity analysis varying (a, b) is recommended to assess robustness.

Minor Points

Correct typographical errors (“generatoring” → “generating”; “Lex X∼” → “Let X∼”).

Ensure consistent notation throughout.

Improve figure captions and add parameter annotations to plots.

Remove editorial artifacts from the manuscript text.

Abstract could briefly mention inclusion of the Power Function distribution as a special case and summarize empirical advantages.

Suggested Additional Analyses and Experiments

Comparative table of model properties to clearly show FPF’s unique position.

Hazard-shape parameter map (α vs. θ) to illustrate full range of hazard behaviors.

Bootstrap-based coverage analysis for MLEs.

Acceptance–rejection acceptance rates and runtime comparisons with inverse transform sampling.

Additional application to a non-survival bounded dataset (e.g., environmental or economic proportion data).

See attachment for Technical Coments and Improvement Guide to Author

Reviewer #2: The manuscript titled “Fréchet–Power Function Distribution: Theory, Properties and Applications” presents a new two-parameter distribution that combines the bounded nature of the Power Function distribution with the heavy-tailed flexibility of the Fréchet generator. The proposed model is mathematically sound, and the derivations of the PDF, CDF, moments, and estimation methods are comprehensive. Simulation studies and applications further highlight its potential usefulness in reliability and survival data analysis.

However, the following issues should be addressed before publication:

1. The introduction lacks a clear justification of why the proposed model is needed, beyond existing extensions. Strengthening motivation would improve the impact.

2. Sections are not properly divided, which makes it harder to follow the flow of the paper. Clearer structuring is recommended.

3. In Section 4 and subsequent parts, captions are written but the graphs themselves are missing. These must be included. In the following sections the same is the case for graphs.

4. Although AIC, BIC, and KS values suggest better fit for the proposed model, the KS test p-value is not significant, which weakens the claim of superior fit. This needs to be discussed more carefully.

5. Include some recent references from 2024 and 2025 to strengthen the literature review and show up-to-date relevance.

6. The English requires editing for grammar and clarity in several places. A thorough language check is recommended.

With these revisions, the paper could make a valuable contribution to the literature on bounded lifetime distributions and data modelling.

**Do you want your identity to be public for this peer review?** For information about this choice, including consent withdrawal, please see our Privacy Policy

Reviewer #1: **Yes: ** Babangida Ibrahim Babura

Reviewer #2: **Yes: ** Shakila Bashir

---

## [Author Response · Author response to Decision Letter 1]

15 Sep 2025

Dear Editor and Reviewers,

We sincerely thank you for the careful review of our manuscript titled “Fréchet–Power Function Distribution: Theory, Properties and Applications.” We appreciate the constructive comments and suggestions, which have greatly helped us to improve the clarity, rigor, and impact of our work.

---

## [Decision Letter · Decision Letter 1]

10 Nov 2025

Dear Dr. Aga,

Thank you for submitting your manuscript to PLOS ONE. After careful consideration, we feel that it has merit but does not fully meet PLOS ONE’s publication criteria as it currently stands. Therefore, we invite you to submit a revised version of the manuscript that addresses the points raised during the review process.

We look forward to receiving your revised manuscript.

Kind regards,

Fucai Lin, Ph.D.

Academic Editor

PLOS ONE

Journal Requirements:

Reviewers' comments:

Reviewer's Responses to Questions

**Comments to the Author**

Reviewer #1: (No Response)

Reviewer #3: (No Response)

2. Is the manuscript technically sound, and do the data support the conclusions?

Reviewer #1: Yes

Reviewer #3: Yes

3. Has the statistical analysis been performed appropriately and rigorously?

Reviewer #1: Yes

Reviewer #3: Yes

4. Have the authors made all data underlying the findings in their manuscript fully available?

Reviewer #1: Yes

Reviewer #3: Yes

5. Is the manuscript presented in an intelligible fashion and written in standard English?

Reviewer #1: Yes

Reviewer #3: Yes

Reviewer #1: The authors have submitted a substantially revised version of their manuscript in response to the first review. The revision is significantly improved in terms of structure, clarity, and depth. Most of the earlier concerns have been carefully addressed: a comparative literature table was added, figures were clarified, hazard and quantile analyses expanded, bootstrap validation and coverage probability analyses included, and the real-data applications were strengthened with explicit goodness-of-fit criteria.

The work now represents a rigorous and well-presented contribution to the statistical literature on bounded heavy-tailed distributions. Only a few issues remain and presented as follows:

1. Include at least one comparative plot overlaying FPF with 2–3 competing bounded models on the same empirical dataset

2. Expand limitations to explicitly cover:

* Potential identifiability issues in regions of (α, θ) that yield similar hazard curves.

* Sensitivity of model fit to misspecification of support boundaries.

* Behavior in small-sample contexts (noting that simulation shows acceptable coverage, but finite-sample bias remains visible).

Reviewer #3: The manuscript titled “Fréchet–Power Function Distribution: Theory, Properties and Applications” presents a new two-parameter model that integrates the characteristics of the Power Function and Fréchet distributions. The author derive several theoretical properties, including the density, cumulative, and quantile functions, as well as moments and hazard rate behavior. They further employ maximum likelihood estimation and simulation techniques for parameter estimation and model validation, with applications to survival and reliability datasets. Overall, the study addresses an important area in statistical modeling and demonstrates commendable effort in developing a flexible and useful distribution for bounded lifetime data. However, I have the following comments that will help improve the quality of the manuscript:

Reference number 5 is incompletely and incorrectly formatted. Please revise it to remove typographical errors and ensure consistency with the journal’s reference style. The corrected form should read as:

Meniconi, M., & Barry, D. M. (1996). The power function distribution: A useful and simple distribution to assess electrical component reliability. Microelectronics Reliability, 36(9), 1207–1212.

The authors are also advised to carefully revise the entire reference section to ensure that all entries are complete and formatted consistently. Please ensure that all references strictly follow the PLOS ONE reference style for uniformity throughout the manuscript.

Please correct the grammatical error in Theorem 1. The sentence should read: “The FPF distribution f(x;θ,α)is a legitimate probability density function.

The authors are encouraged to update the reference list by including the following relevant papers, which are closely related to the present study:

1. A Study on Topp-Leone Kumaraswamy Fréchet Distribution with Applications: Methodological Study.

2. Modeling of COVID-19 Datasets with Three-Parameter Fretchet Distribution

3. A hybrid cosine inverse Lomax-G family of distributions with applications in medical and engineering data.

It is suggested that the authors include a brief paragraph introducing the relevance of the survival and hazard functions before presenting their mathematical formulations.

In the application section, on the second line after presenting the CDF of the seventh competing model, the citation “McGilchrist C. and Aisbett C. [31]” should be corrected. There is no need to include the initials since the authors are cited within the text. Please remove the initials “C.” and retain only the last names, i.e., “McGilchrist and Aisbett [31].

The authors should consider including a brief discussion on the limitations of the study and possible directions for future research in the conclusion section. This addition will provide a clearer perspective on the scope of the current work and potential extensions.

**Do you want your identity to be public for this peer review?**  For information about this choice, including consent withdrawal, please see our Privacy Policy

Reviewer #1: **Yes: ** Babangida Ibrahim Babura

Reviewer #3: No

---

## [Author Response · Author response to Decision Letter 2]

10 Nov 2025

We sincerely thank the editor and reviewers for their constructive and insightful comments, which have greatly improved the quality and clarity of our manuscript. All suggestions have been carefully considered and incorporated as follows:

---

## [Decision Letter · Decision Letter 2]

13 Nov 2025

Frechet-Power Function Distribution: Theory, Properties and Applications

PONE-D-25-39356R2

Dear Dr. Aga,

We’re pleased to inform you that your manuscript has been judged scientifically suitable for publication and will be formally accepted for publication once it meets all outstanding technical requirements.

Kind regards,

Fucai Lin, Ph.D.

Academic Editor

PLOS ONE

Reviewer #1: The manuscript is technically sound, and the data support the conclusions.

The statistical analysis where performed appropriately with rigor.

The authors made all data underlying the findings in their manuscript fully available.

The manuscript presented in an intelligible fashion and written in standard English.

Reviewer #3: (No Response)

---

## [Editor Report · Acceptance letter]

PONE-D-25-39356R2

PLOS ONE

Dear Dr. Aga,

I'm pleased to inform you that your manuscript has been deemed suitable for publication in PLOS ONE. Congratulations! Your manuscript is now being handed over to our production team.

Kind regards,

on behalf of

Professor Fucai Lin

Academic Editor

PLOS ONE